# Layer-by-layer biofunctionalization of nanostructured porous silicon for high-sensitivity and high-selectivity label-free affinity biosensing

Stefano Mariani[1], Valentina Robbiano [1], Lucanos M. Strambini[2], Aline Debrassi[3], Gabriela Egri[3], Lars Dähne[3] & Giuseppe Barillaro [1,2]

Nanostructured materials premise to revolutionize the label-free biosensing of analytes for clinical applications, leveraging the deeper interaction between materials and analytes with comparable size. However, when the characteristic dimension of the materials reduces to the nanoscale, the surface functionalization for the binding of bioreceptors becomes a complex issue that can affect the performance of label-free biosensors. Here we report on an effective and robust route for surface biofunctionalization of nanostructured materials based on the layer-by-layer (LbL) electrostatic nano-assembly of oppositely-charged polyelectrolytes, which are engineered with bioreceptors to enable label-free detection of target analytes. LbL biofunctionalization is demonstrated using nanostructured porous silicon (PSi) inter-ferometers for affinity detection of streptavidin in saliva, through LbL nano-assembly of a bi-layer of positively-charged poly(allylamine hydrochloride) (PAH) and negatively-charged biotinylated poly(methacrylic acid) (b-PMAA). High sensitivity in streptavidin detection is achieved, with high selectivity and stability, down to a detection limit of 600 fM.

[1] Dipartimento di Ingegneria dell'Informazione, Università di Pisa, Via G. Caruso 16, 56122 Pisa, Italy. [2] Istituto di Elettronica e di Ingegneria dell'Informazione e delle Telecomunicazioni, Consiglio Nazionale delle Ricerche, Via G. Caruso 16, 56122 Pisa, Italy. [3] Surflay Nanotec GmbH, Max-Planck-Straße 3, 12489 Berlin, Germany. Correspondence and requests for materials should be addressed to G.B. (email: giuseppe.barillaro@unipi.it)

Surface biofunctionalization plays a pivotal role in biosensing, when either electrical or optical transducers are exploited, as it imparts to the transducer all the necessary features for the selective and sensitive detection of the target analyte. It consists of two chief steps, namely, physico-chemical surface activation and bioreceptor immobilization, both of which have a tremendous effect on selectivity and sensitivity of the resulting biosensor[1]. In fact, yield and stability of the different chemical sub-steps of both surface activation and bioreceptor immobilization processes regulates the number of bioreceptors available at the transducer surface for unit area (bioreceptor density) and over time for the biorecognition of the target analyte. Besides, the bioreceptor orientation might also play a role, particularly for affinity biosensing, in setting the bioreceptor density on the surface of the transducer and, in turn, the specificity/sensitivity of the biomolecular recognition process[2].

For instance, if we focus the attention to biosensing with optical platforms exploiting silicon-derivative (e.g., Si, $SiO_2$, $SiO_x$) transducers, the surface activation of the transducer is mainly carried out through either organosilanization of an oxidized silicon surface, which leads to the formation of polar, covalent Si-O-Si bonds between the surface and organosilane molecules, or by direct hydrosilylation of Si-H surfaces, which results in a self-assembled monolayer (SAM) of alkyl chains anchored to the surface through non-polar, covalent Si-C bonds[3,4].

Organosilanization undoubtedly represents an attractive approach, being quite straightforward and relatively cheap. However, the Si-O-Si bond at the surface is inherently prone to hydrolysis in aqueous media[3,4] and formation of multilayers is likely to occur due to physisorption of organosilanes onto the surface[5]. Both these issues might lead to a progressive change of the bioreceptor density at the transducer surface over time, which negatively impacts efficiency, stability, and reproducibility of the whole biofunctionalization process.

On the other hand, the Si-C bond achieved through hydrosilylation of alkenes and alkynes[4,5] features a good stability also in extreme conditions (e.g., boiling KOH solution, pH = 12)[6], thus providing a very attractive alternative to organosilanization. However, the metastability of the native Si-H surfaces, which are prone to oxidation in environmental conditions, and, in turn, the need of performing the hydrosilylation reaction in an inert, deoxygenated, and humidity-free atmosphere, has prevented the popularization of this approach for biosensing.

Generally speaking, the density of bioreceptors available at the transducers surface is set by both yield $\gamma_n$ and number $n$ of the chemical steps needed to activate the surface and secure the bonding of the bioreceptor molecules, where the value of $\gamma_n$ is always < 1 (i.e., < 100%) for real processes. Therefore, the yield of the entire biofunctionalization process $\gamma_{tot}$ might be relatively low already on flat surfaces being $\gamma_{tot} = \prod_1^n \gamma_n$, and it is expected to be significantly lower on nanostructured surfaces, where issues related to diffusion, steric demand, and orientation of molecules plays a major role[3,4].

The layer-by-layer (LbL) assembly is a pervasive method for conformal surface coating of substrates with polymers, colloids, biomolecules, and cells, which offers superior control and versatility with respect to other thin-film deposition methods, especially on micro and nanostructured surfaces[7].

Conventional LbL assembly was initially reported by Decher in 1997[8], who firstly demonstrated the formation of multilayer architectures by sequentially adsorbing oppositely charged polyelectrolytes (i.e., polyanions and polycations) onto a substrate, through exploitation of enthalpic and entropic driving forces. As

the technique gained interest, a range of interactions, such as hydrophobic interactions[9], hydrogen bonding[10], and covalent coupling[11], have been later exploited to prepare multilayered films via LbL assembly. Over the past 20 years, LbL assembly has been successfully employed for many different applications, from separation science[12,13] to drug delivery[14–23], from biomedicine[24,25] to biosensing[26–29].

As to biosensing, LbL assembly has been mostly used for electrochemical/enzymatic detection of (bio)molecules. LbL-based electrochemical/enzymatic biosensors with improved analytical performance have been reported, thanks to electrostatic and massive capturing of bioreceptors on charged LbL-coated surfaces, which does not affect the native conformation of enzymes, as well as incorporation of metallic nanomaterials in the polyelectrolytes, which improves the electronic communication between enzymes and electrodes[26].

To our best knowledge, application of LbL assembly to label-free and affinity optical biosensing has been totally overlooked so far, because of severe selectivity problems in detecting specific binding events of the target analyte with bioreceptors entrapped in the LbL assembly. This is mostly due to the electrostatic unspecific adsorption of both target and interfering biomolecules onto the charged LbL layers. As a matter of fact, the only two works that have been reported to date on the use of LbL assembly for label-free optical biosensing rely on the electrostatic and unspecific immobilization of bioreceptors onto the charged polyelectrolytes of the LbL assembly, either IgG antibodies for the anti-IgG detection with a fiber-optic Fabry-Perot interferometer[28] or aptamer-probes for C-reactive protein (CRP) detection with a lossy-mode-resonance optical fiber[29]. In both cases, no data were provided to rule out unspecific electrostatic interaction between target analytes and charged polyelectrolytes without bioreceptors.

Among the different materials that have been used for the preparation of label-free optical biosensors, nanostructured porous silicon (PSi) has been increasingly exploited due to its huge specific surface, which allows a tremendous number of molecules to be accommodated, straightforward fabrication, which allows high versatility in preparation of optical structures to be achieved, and low cost, which allows mass production of cheap biosensors for point-of-care application to be envisaged[30]. Nonetheless, limit of detection (LoD) of label-free PSi-based biosensors is bound to µM–nM concentrations, in spite of transducer architectures (e.g., interferometrs[31–33], resonant microcavities[34,35], rugate filters[36]), surface chemistry (e.g., organosilanization[32], hydrosililation[36]), bioreceptors (e.g., antibody, aptamer), assay strategy (e.g., preconcentration of the target analyte[31] recirculation of the target analyte[32]), and readout technique (e.g., Fast Fourier Transform (FFT)[30], Interferogram Average over Wavelength (IAW)[33]). This can be ascribed to poor yield and reliability of the multi-steps covalent chemistry functionalization of nanopores (diameter 50 nm) with very high aspect ratio (about 100).

In this work, we report on the surface biofunctionalization of nanostructured materials via LbL nano-assembly as a robust and effective alternative route to standard covalent chemistry, e.g., organosilanization and hydrosililation, for the development of a class of affinity biosensors with superior performance.

Surface biofunctionalization via LbL nano-assembly leverages the electrostatic self-assembly of oppositely charged polyelectrolytes, which were engineered to carry bioreceptors covalently bound to the polymer chain, onto the transducer surface. This enables tackling the main drawbacks of covalent chemistry when carried out on nanostructured surfaces, namely, poor control of molecule nanolayer assembling on the surface, scarce

reproducibility of the different biofunctionalization steps, low stability of the bioreceptor immobilization on the surface, low yield of the overall biofunctionalization process, so improving the analytical performance of LbL-biofunctionalized biosensors, in terms of sensitivity, stability, and reproducibility. Further, LbL biofunctionalization allows selectivity of the bioreceptor-target binding event to be further boosted by enabling an electrostatic repulsion between charged LbL assembly and charged non-target proteins unspecifically trapped in the LbL assembly. This is achieved by setting up a repulsive rinsing step at a pH value significantly different from pI values of both target and non-target proteins, so as to make all proteins charged with the same polarity of the outer layer of the LbL nano-assembly.

A proof-of-concept demonstration of the superior biosensing performance of LbL biofunctionalization over silane-based covalent chemistry is here given using a nanostructured PSi interferometer for label-free biotin-streptavidin affinity biosensing.

LbL biofunctionalization of PSi interferometers enables the preparation of stable and reliable label-free optical biosensors with a LoD for streptavidin of 600 fM, which is about five orders of magnitude (i.e., $10^5$) lower than that achieved on control PSi interferometers prepared with silane-based chemistry (about 100 nM). This also represents a 300-fold improvement with respect to the state-of-the-art on PSi biosensors, both in terms of LoD and sensitivity, pushing analytical performance of label-free PSi biosensors to that of the most sensitive label-free biosensing platforms, namely surface plasmon resonance (SPR), local surface plasmon resonance (LSPR), interferometers, ring resonators, photonic crystals, optical fibers, as highlighted in Fig. 1.

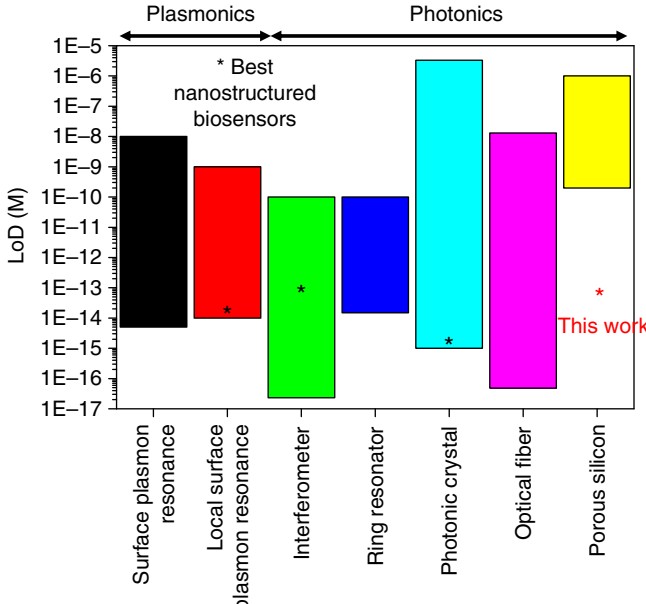

**Fig. 1** State-of-art plasmonic and photonic platforms for high-sensitivity label-free biosensing. For each platform, namely SPR[54], LSPR[55], interferometer[56,57], ring resonator[58], photonic crystal[59], optical fiber[60], the best-performing biosensor exploiting nanostructured materials is indicated with an asterisk, when available. Porous silicon (PSi) nanostructured biosensors performed less well than the other platforms to date, at least in terms of sensitivity and, in turn, limit of detection (LoD). The LbL biofunctionalization of nanostructured materials proposed in this work allows the gap between PSi optical biosensors and other optical biosensing platforms to be filled, thus pushing PSi biosensors to minimum detectable analyte concentrations comparable to those of the best-performing label-free biosensing platforms

Remarkably, stability and efficacy of LbL biofunctionalization was successfully confirmed in a complex and non-filtered body fluid spiked with streptavidin, namely saliva, which is a largely available and easily accessible non-invasive body fluid of increasing interest for bioclinical assays.

## Results

**LbL biofunctionalization of PSi interferometers.** Electro-statically driven LbL biofunctionalization of PSi interferometers for biotin/streptavidin affinity biosensing was carried out as sketched in Fig. 2a. Oxidized PSi interferometers were prepared through a two-step electrochemical etching of crystalline silicon wafers and subsequent thermal oxidation (Fig. 2a-1). A first PSi sacrificial layer was produced through anodic etching and immediately dissolved by alkaline etching to provide the silicon surface with a nanostructured texture with tiles of average size of 50 nm[33]. A PSi interferometer with thickness of ~5 μm, porosity of ~80%, and pore size of ~50 nm (aspect ratio of 100) was next produced through anodic etching of the so-textured silicon sub-strate (Figs. 2b, c and Supplementary Figure 2a, b). The as-prepared PSi interferometer was subjected to thermal oxidation to convert silicon to silicon dioxide, so as to achieve a hydrophilic, negatively charged surface that was functional to carry out an electrostatically driven LbL coating of the PSi surface. Drop casting of a positively charged polyelectrolyte, namely poly(ally-lamine hydrochloride) (PAH), solution onto the oxidized PSi interferometer, resulted in the conformal electrostatic adsorption of a PAH nanolayer (a few nm[37,38]) onto the inner surface of the nanopores (Fig. 2a-2). Eventually, drop casting of a biotinylated negatively charged polyelectrolyte, namely biotinylated poly (methacrylic acid) (b-PMAA), solution onto the PAH-coated PSi interferometer, resulted in the conformal electrostatic adsorption of a b-PMAA nanolayer onto the PAH, thus achieving a single bi-layer of PAH/b-PMAA with bioreceptors directly bonded to the external PMAA polyelectrolyte (Fig. 2a-3).

Optical characterization of the PSi interferometers through UV−Vis FFT reflectance spectroscopy (FFT-RS) showed a consistent variation of the optical thickness (effective optical thickness (EOT) = $2n_{eff}t$, with $n_{eff}$ effective refractive index and $t$ thickness of the PSi interferometer) at the different preparation steps (Supplementary Figure 1c,d). The results are summarized in Fig. 1d. Specifically, the EOT values of as-prepared PSi interferometers (13,934 ± 490 nm, calculated from reflectance spectra acquired in air) decreased after thermal oxidation (reduction of −1177 ± 160 nm) due to partial conversion of silicon to silicon dioxide, then consistently increased (with respect to oxidized PSi interferometers) upon electrostatic LbL-coating of PAH (111 ± 50 nm) and b-PMAA (390 ± 157 nm). The effective refractive index ($n_{eff}$) of the PSi interferometers (average thickness of 4.92 μm) lowered from 1.416 Refractive Index Unit (RIU) for the as-prepared PSi interferometer to 1.296 RIU after oxidation, and then increased to 1.301 RIU after PAH coating and 1.332 RIU after b-PMAA coating.

To check homogeneity of LbL assembly over the entire depth of PSi interferometers, we infiltrated an oxidized PSi interferometer with a sulfo-rhodamine-labeled PAH (a red-emitting fluorescent dye), as well as a PSi interferometer previously coated with non-labeled PAH with a fluorescein-labeled PMAA (a green-emitting fluorescent dye), using the deposition protocol previously described. Figures 2e-1 and 2e-2 show bright-field and fluorescence optical microscopy images, respectively, of the cross-section of a PSi interferometer infiltrated with sulfo-rhodamine-labeled PAH; Fig. 2e-3 shows the cross-section of a PSi interferometer, previously coated with non-labeled PAH, after infiltration of fluorescein-labeled PMAA. Negative control

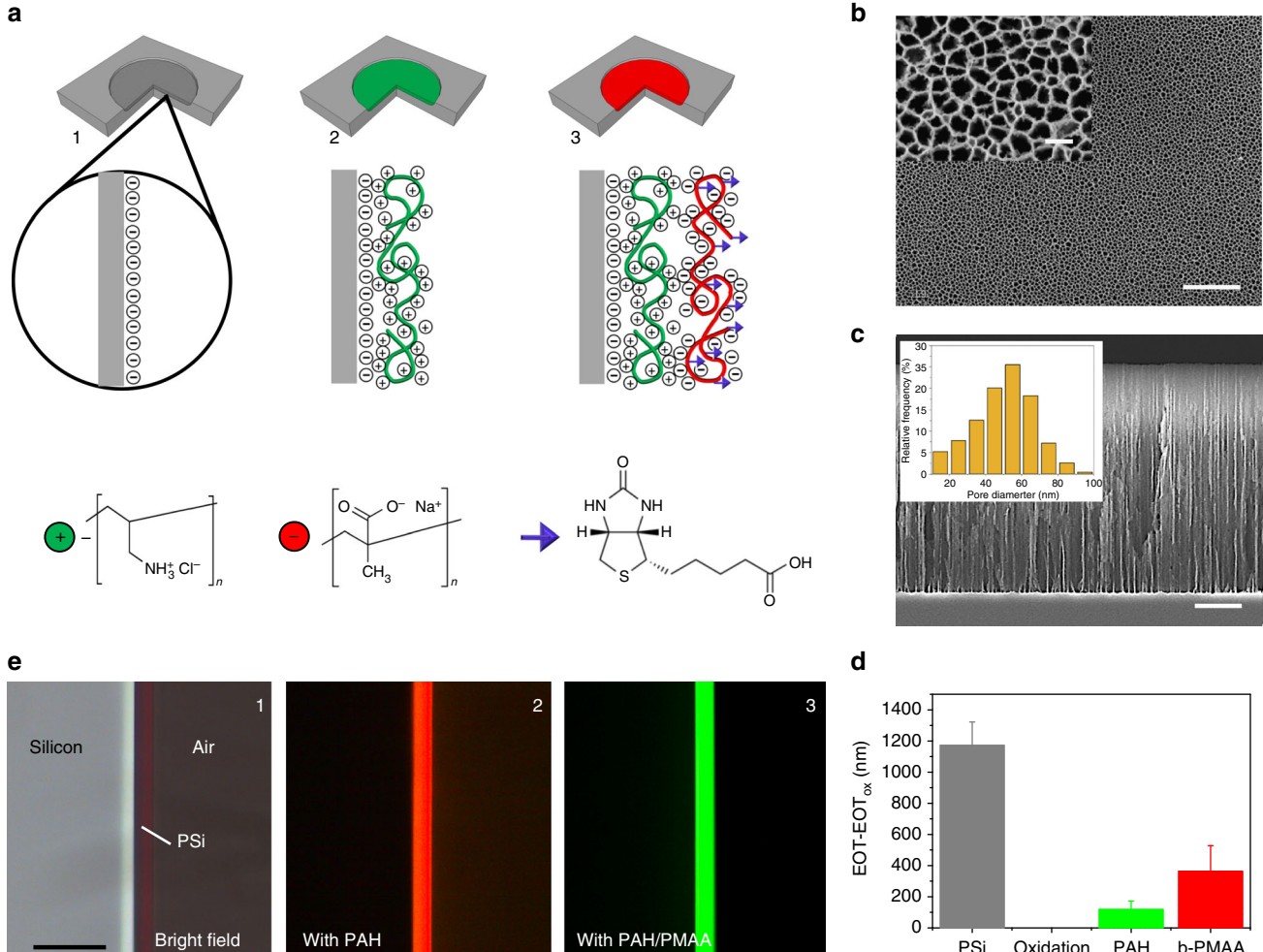

**Fig. 2** Layer-by-Layer biofunctionalization of nanostructured PSi interferometers. **a** Sketch of the biofunctionalization of the nanostructured surface of PSi interferometers via LbL nano-assembly: (1) preparation and oxidation of PSi interferometers (gray, negatively charged); (2) PAH coating (green, positively charged) of oxidized PSi interferometers; (3) b-PMAA (biotinylated PMMA) coating (red, negatively charged) of PSi/PAH interferometers. Details of the chemical structure of the polyelectrolytes used for LbL assembly, namely PAH (green circle) and PMAA (red circle), and of biotin (violet triangle) covalently linked to PMAA by amino-coupling, are given at the bottom of **a**. **b** SEM top-view image (50,000 × magnification) of the PSi interferometer surface (scale bar is 1.00 μm). The inset shows a higher magnification SEM image (200,000 × magnification) of the PSi surface that allows pore arrangement and size to be clearly appreciated (scale bar is 100 nm). **c** SEM cross-section image (25000 × magnification) of the PSi interferometer highlighting the columnar structure of the pores (scale bar is 1.00 μm). The inset shows a histogram of the pore size distribution, highlighting an average diameter of about 55 nm. **d** EOT-EOT$_{ox}$ values recorded for PSi interferometers both as-prepared and after each LbL biofunctionalization step. The EOT value of oxidized PSi interferometers (i.e., EOT$_{ox}$) is used as reference to obtain positive differential EOT values. Data are provided as average values over seven replicates with error bars representing one standard deviation. **e** Bright-field optical image (1) and fluorescence images (2, 3) of the cross-section of oxidized PSi interferometers LbL-coated with (2) sulfo-rhodamine-labeled PAH and with (3) fluorescein-labeled PMAA on top of non-labeled PAH coating (scale bar is 15 μm). The homogeneity of the electrostatically driven LbL coating can be easily appreciated by the uniform fluorescence emission of the labeled polyelectrolytes over the whole PSi thickness

fluorescence images of oxidized PSi interferometers both bare (Supplementary Figure 2a, b) and coated with PAH/PMAA (Supplementary Figure 2c, d) are reported as Supplementary Information. From fluorescence images, it is apparent that the pores are uniformly covered with both the polyelectrolytes all over their depth, in spite of their nanometric diameter (50 nm in average) and high aspect ratio (about 100).

A chief advantage of LbL biofunctionalization via charged polyelectrolytes, with respect to covalent chemistry, concerns the electrically induced self-regulation of the thickness of both single and multilayer films, which turns into a superior homogeneity and reproducibility of LbL films when deposited within high aspect ratio nanostructures (e.g., nanopores), where constrained diffusion of molecules might lead to a reduction of the

concentration with depth and, in turn, to non-homogeneous surface coating[37]. The thickness is few nm per layer regardless of the substrate used and increases linearly with the number of layers deposited, which makes the film properties highly controllable and rather independent of the substrate. The polyelectrolyte thickness and conformation at the surface and, in turn, the newly created film surface is thus mostly dependent on the chosen polyelectrolytes and adsorption conditions, and less dependent on the substrate or the substrate charge density[38].

Moreover, the use of polyelectrolytes for surface biofunctionalization, rather than organo-linkers as in covalent chemistry, is advantageous in terms of good adhesion of the nanolayer to the underlying substrate, because polymers can bridge over underlying defects. In fact, the electrostatic attraction between

oppositely charged molecules for multilayer formation has shown to have the least steric demand of all chemical bonds[24]. Conversely, covalent chemistry biofunctionalization (either silanization or hydrosylilation) is restricted to certain classes of organics that give rise to self-assembled multilayer architectures, e.g., organo-linker/linking-molecule/bioreceptor, of poor quality and with scarce reliability[3,4]. These problems are likely caused by the high steric demand of covalent chemistry and the severely limited number of reactions with exactly 100% yield, which is a prerequisite for the preservation of functional group density in each layer[24].

A further major advantage of LbL biofunctionalization from polyelectrolyte solution is that different building-blocks, e.g., bioreceptors, can be incorporated in individual films and that difficult-to-control (unreliable) chemical steps, e.g., coupling of bioreceptors, can be determined in solution, before going to the nanostructured surface. In addition, the multilayer architecture is completely determined by the deposition sequence[24]. This leads to a controlled and uniform distribution of bioreceptors over the coated surface, which would allow both design and performance of biosensors built using LbL nano-assembly to be significantly improved with respect to those achieved with the standard chemistry counterpart.

**Stability of LbL assembly with pH and ionic strength**. A major issue when dealing with surface biofunctionalization for sensing purposes is reliability and stability of the multilayer film deposited on the surface itself, which, if poor, would result in an inefficient surface coverage and loss of bioreceptors[3,4].

Stability of LbL assembly was assessed in the presence of three different buffers, commonly used for bioassays and featuring different pH and ionic strength $I$ values, namely, acetate buffer (10 mM $CH_3COOH/CH_3COONa$ with 100 mM NaCl, $I = 0.106$ M, pH = 5); phosphate-buffered saline (PBS) buffer (100 mM $NaH_2PO_4/Na_2HPO_4$ with 100 mM NaCl, $I = 0.36$, pH = 7.4); and, HEPES buffer (10 mM with HEPES with 100 mM NaCl, $I = 0.103$, pH = 7.4).

The LbL-biofunctionalized PSi interferometers were secured into a flow cell and stability was investigated through FFT reflectance spectroscopy, which allows refractive index variation due to deterioration of the LbL assembly to be reliably detected. The EOT value was continuously monitored over time for 80 min (at least) upon infiltration of the flow cell with the buffers at 100 $\mu L\ min^{-1}$.

Excellent stability of the EOT signal over time was recorded in acetate buffer (EOT-$EOT_0 = -3.8 \pm 12.6$ nm, average value and standard deviation over 80 min, where $EOT_0$ is the reference value at $t = 0$) (Supplementary Figure 3a), clearly demonstrating that the LbL assembly is very stable at such pH and ionic strength conditions. Infiltration of HEPES buffer (same ionic strength and higher pH compared with acetate), after stabilization in acetate buffer, led to a small variation of the EOT signal that is compatible with the refractive index change between HEPES and acetate buffers (Fig. 3a). Remarkably, the EOT signal was well stable in HEPES (EOT-$EOT_0 = -11.9 \pm 1.6$ nm, average value and standard deviation over 30 min) and went back to the baseline value once acetate was infiltrated back in the flow cell. This clearly indicates that the LbL assembly is stable at such a higher pH value (namely 7.4), at low ionic strength. Eventually, infiltration of PBS buffer in the flow cell, after stabilization in acetate buffer, led to an abrupt, major blue-shift of the EOT signal, with respect to the value in acetate, which is not compatible with the variation of the refractive index between PBS and acetate buffers, followed by an important drift over time (Supplementary Figure 3b). Once acetate buffer was further

infiltrated in the flow cell the EOT signal did not go back to the baseline value, thus indicating that an irreversible change of the LbL assembly occurred in PBS. We argue that the higher ionic strength of PBS ($I = 0.36$), with respect to that of both acetate and HEPES buffers (about 0.1), weakens the electrostatic interactions between $SiO_2$, PAH, and PMAA charged layers, thus promoting the LbL multilayer de-assembly. Figure 3b summarizes the results on the stability of the LbL nano-assembly in acetate, PBS, and HEPES buffers. The disruption of the LbL assembly in PBS is apparent from the strong and irreversible change of the EOT value after infiltration of PBS in the LbL-coated PSi interferometers.

**Optimizing specific/unspecific binding with repulsive rinsing**. In spite of the extensive use of electrostatic LbL assembly for electrochemical/enzymatic biosensing, its use in affinity optical and affinity biosensing is still in its infancy[26,27]. In fact, the intrinsic unspecific interaction between charged coating layers and oppositely charged either target bioanalytes (in the absence of bioreceptors) or interfering analytes (in the presence of bioreceptors) has posed severe limitations so far.

The minimization of unspecific adsorption is here successfully achieved leveraging a rinsing step that ensures electrostatic repulsion between the LbL assembly and both target analyte and interfering biomolecules that are not specifically bound to the bioreceptors. The repulsive rinsing step ensures that only the target analytes that are specifically bound to their biorepceptors will be retained after rinsing. Conversely, any of the analytes (both target and interfering) that are unspecifically trapped in the outer polyelectrolyte layer of the LbL assembly will be removed during the rinsing step by repulsive electrostatic forces established between polyelectrolyte and biomolecules electrically charged with same polarity. Figures 3c, e show a sketch of specific target binding and unspecific repulsive rinsing for biofunctionalized and control LbL-coated interferometers, respectively.

To assess minimization of unspecific adsorption through repulsive rinsing, we prepared PSi interferometers coated with both PAH and either biotinylated b-PMAA (biotin:PMAA monomer = 1:30) or non-biotinylated PMAA (as control). Notice that the outer PMAA layer of the LbL assembly is negatively charged regardless of the biotinylation. The target analyte was streptavidin, and interfering biomolecules taken into account were pepsin and bovine serum albumin (BSA). Streptavidin is globally neutral at pH = $pI_{strept} = 5.0$ and deeply shielded by a relatively high ionic strength (NaCl 100 mM). To minimize protein–protein ionic repulsion and improve, in turn, streptavidin diffusion inside the nanopores[39], we used acetate buffer (pH = 5.0) with 100 mM of NaCl as running buffer. On the other hand, BSA and pepsin have an isoelectric point of $pI_{BSA} = 4.7$ and $pI_{pepsin} = 1$, respectively. Therefore, in order to yield both target and interfering biomolecules negatively charged in the rinsing step and enable, in turn, electrostatic repulsion between unspecifically bound biomolecules and PMAA, we used HEPES (pH = 7.4 > $pI_{BSA}$, $pI_{pepsin}$, $pI_{strept}$) as a rinsing buffer.

The IAW reflectance spectroscopy (IAWRS) was used for the real-time monitoring of the interactions (specific and unspecific) of both b-PMAA and PMAA with streptavidin, pepsin, and BSA. The IAWRS was here preferred to FFT-RS as the former guarantees both higher sensitivity and greater reproducibility with respect to the latter[33,40], enabling the reliable measurement of concentrations below the nM level, which would not be otherwise appreciable using FFT-RS.

Acetate was initially injected in the flow cell for 40 min to ensure IAW signal stabilization (Phase 1), then streptavidin at 500 $\mu g\ mL^{-1}$ (i.e., ~ 8.3 $\mu$M) was infiltrated at 5 $\mu L\ min^{-1}$ for 40

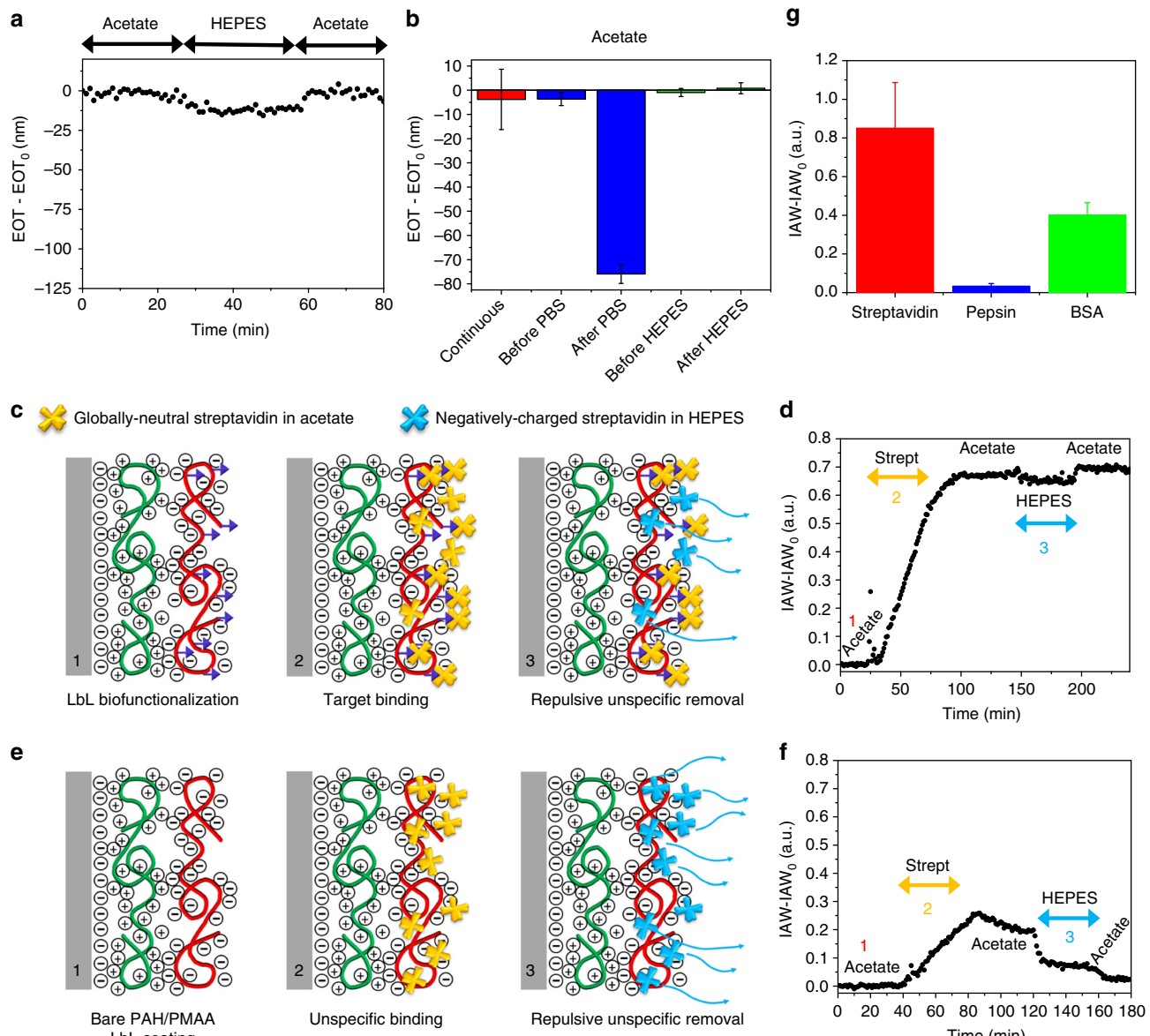

**Fig. 3** Stability and specificity of LbL biofunctionalization. **a** Sensorgram (EOT-EOT$_0$ vs. time) recorded on LbL-biofunctionalized PSi interferometers highlighting high stability of the LbL nano-assembly both in acetate and HEPES buffers. **b** EOT-EOT$_0$ values acquired on LbL-biofunctionalized PSi interferometers upon continuous infiltration of acetate buffer, as well as in acetate buffer before and after injection of PBS and HEPES buffers. Data are provided as average values of EOT-EOT$_0$ fluctuations over time (data points >30) with error bars representing one standard deviation. **c** Sketch of the inner surface of PSi interferometers LbL-biofunctionalized with PAH/b-PMAA (1), highlighting affinity (specific) binding of streptavidin with biotin (2) and repulsive rinsing of streptavidin unspecifically adsorbed (electrostatic interaction) on b-PMAA (3). **d** Sensorgram (IAW-IAW$_0$ vs. time) acquired on LbL-biofunctionalized PSi interferometers upon injection of streptavidin 8.3 μM: signal stabilization in acetate buffer (step 1); injection of streptavidin and affinity interaction of streptavidin with b-PMAA (step 2); repulsive rinsing in HEPES buffer and removal of streptavidin unspecifically adsorbed on b-PMAA (step 3). Acetate buffer is eventually injected to allow a direct comparison of the IAW values before streptavidin injection and after repulsive rinsing to be feasible. **e** Sketch of the inner surface of PSi interferometers LbL-coated with bare PAH/PMAA (non-biotinylated) (1), showing unspecific interaction of streptavidin with PMAA (2) and complete removal of streptavidin unspecifically adsorbed on PMAA by repulsive rinsing (3). **f** Sensorgram (IAW-IAW$_0$ vs. time) acquired on PSi interferometers LbL-coated with bare PAH/PMAA (non-biotinylated) upon injection of streptavidin 8.3 μM (steps 1 through 3 as in **d**). **g** IAW-IAW$_0$ values measured on LbL-biofunctionalized PSi interferometers upon injection of 8.3 μM streptavidin (target molecule), as well as 14 μM Pepsin and 7.6 μM BSA (non-target molecules), highlighting that a high specificity is achieved through repulsive rinsing. Data are provided as average values over 3 replicates with error bars representing one standard deviation

min (Phase 2). The IAW signal increased to ~0.7 a.u., with respect to the IAW$_0$ reference value in acetate, for the LbL-biofunctionalized interferometers (Fig. 3d), whereas for the control interferometers it only increased to ~0.25 a.u. (Fig. 3f). We argue that the 3× increase in the IAW variation of biofunctionalized interferometers, with respect to control

interferometers, is a clear indication that streptavidin–biotin affinity binding occurred in the former. On the other hand, unspecific adsorption of streptavidin in the PMAA network is responsible for the signal increase in the latter. In fact, the rinsing step in acetate (40 min at 100 μL min$^{-1}$) clearly shows that the IAW signal of biofunctionalized interferometers stabilizes

without appreciable reduction over time, whereas a slight decrease is observed for control interferometers due to partial removal of streptavidin unspecifically trapped in the PMAA layer. The next repulsive rinsing step in HEPES (40 min at 100 $\mu$L min$^{-1}$, Phase 3) fully removes streptavidin that is unspecifically bound to the PMAA network through electrostatic repulsion of proteins now charged with same polarity as the PMAA layer. This is apparent for control interferometers (Fig. 3f), where the IAW signal quickly and significantly decreases after HEPES injection, thus demonstrating that the repulsive rinsing allows most of the streptavidin unspecifically bound to PMAA to be removed. Remarkably, by further injecting acetate (reference buffer, 40 min at 100 $\mu$L min$^{-1}$) it is feasible to directly compare (i.e., in the same buffer) IAW values before streptavidin injection and after repulsive rinsing, from which a negligible residual unspecific signal is visible for control interferometers, namely IAW-IAW$_0$ = 0.028 $\pm$ 0.005 a.u over three replicates. On the other hand, for the biofunctionalized interferometers the IAW-IAW$_0$ value does not significantly change after rinsing, confirming that the streptavidin is specifically bound to bioreceptors in this case (Fig. 3d). Notice that, the change of the IAW value with HEPES in Figure 3d can be mainly ascribed to a change in the refractive index between HEPES and acetate buffers.

To further corroborate the validity of the repulsing rinsing step in HEPES, unspecific interaction between the LbL assembly (PAH/PMAA) and non-target proteins, namely pepsin and BSA, was investigated. Notice that BSA represents a worst case, being BSA a protein commonly employed in surface passivation due its ability to effectively and unspecifically adsorb to materials (even nanostructured). The results are summarized in Fig. 3g. Injection of pepsin and BSA at 500 $\mu$g mL$^{-1}$ (i.e., ~ 14 $\mu$M for pepsin and ~7.6 $\mu$M for BSA) resulted in an IAW-IAW$_0$ signal of 0.033 $\pm$ 0.014 and 0.401 $\pm$ 0.064 a.u., respectively (Supplementary Figure 3c,d), both of which are significantly smaller than the value recorded for the specific biding of streptavidin (i.e., 0.8 a.u.). Remarkably, pepsin is almost fully removed from PMAA (specificity ratio between streptavidin and pepsin ~ 35 and IAW-IAW$_0$ comparable to the value recorded for the selective binding of streptavidin 8.3 pM, i.e., 0.031 $\pm$ 0.006 a.u., see next paragraph). Moreover, in spite of the relatively high IAW-IAW$_0$ value achieved for BSA, the sensorgram recorded for BSA (Supplementary Figure 3d) clearly shows that the removal efficiency of BSA unspecifically adsorbed on PMAA is nearly 60%, which is roughly the same of that achieved using a denaturation solution of 0.5% sodium dodecyl sulfate in aqueous solution[41].

**Biosensing with LbL-biofunctionalized PSi interferometers.** Three PMAA polyelectrolytes with different degrees of biotinylation, namely, biotin:PMAA monomer = 1:65, 1:40, and 1:30, were prepared to investigate how the biotinylation degree impacts the sensing performance of PSi interferometers. Regardless of the biotinylation degree, specific binding of streptavidin at concentration of 8.3 $\mu$M (i.e., 500 $\mu$g mL$^{-1}$) was tested according to the injection/rinsing protocol described above. Figure 4a summarizes the IAW-IAW$_0$ values obtained upon injection of streptavidin 8.3 $\mu$M using PMAA with different biotinylation degrees; nonspecific IAW-IAW$_0$ value recorded at the same streptavidin concentration for bare (i.e., non-biotinylated) PMAA is also reported. The sensorgrams of the IAW-IAW$_0$ signals recorded for the different biotinylation degrees are provided in Fig. 3d for PMAA 1:30, and in Supplementary Figure 4a,b for PMAA 1:65 and 1:40, respectively. The IAW-IAW$_0$ signal, due to specific binding of streptavidin and biotin, recorded for b-PMAA at the smallest biotinylation degree is one order of magnitude

higher (10×) than that recorded for bare PMAA. Moreover, it increases significantly with the degree of biotinylation of the PMAA (Fig. 4a, Supplementary Figure 4c). These results envisage the possibility of further increasing the IAW-IAW$_0$ output signal of LbL-biofunctionalized biosensors by increasing the biotinylation degree of PMAA, to some extent at least.

Evaluation of the analytical performance of LbL-biofunctionalized PSi interferometers was carried out using PMAA with biotinylation degree 1:30, at streptavidin concentrations ranging from 0.5 ng mL$^{-1}$ (~ 8.3 pM) to 500 $\mu$g mL$^{-1}$ (~8.3 $\mu$M).

Figure 4b shows the calibration curve, namely, IAW-IAW$_0$ vs. streptavidin concentration in the range 8.3 pM to 8.3 $\mu$M, achieved over three different PSi interferometers functionalized with b-PMAA(1:30) (red dots, log-log scale). A magnification of the calibration curve in the region 8.3 pM to 8.3 nM is reported in Fig. 4c in linear scale, where the higher sensitivity of the interferometer at the smaller concentrations, with respect to higher ones, can be better appreciated. The gray area indicates the region were the IAW-IAW$_0$ signal is below $3.3\sigma_{IAW0}$ value, being $\sigma_{IAW0}$ = 0.004 a.u. the standard deviation of the IAW$_0$ reference signal (no streptavidin) measured over three replicates. A power-law trend encompassing the whole range of concentrations tested is apparent from Fig. 4b, which is best-fitted ($R^2$ = 0.98) with the following equation (red solid trace):

$$IAW - IAW_0 = 0.070 \times C^{0.227} \qquad (1)$$

where $C$ is the concentration of the streptavidin expressed in nM.

All the tested concentrations are well discriminated with respect to the noise floor with excellent signal-to-noise ratio (SNR), namely, SNR = 212 for 8.3 $\mu$M and 7.5 for 8.3 pM. Figure 4d shows the typical sensorgram (raw data) acquired for the lowest tested streptavidin concentration (i.e., 8.3 pM), highlighting that a stable signal can be measured at such a small concentration over the noise level. The biosensor reproducibility evaluated over the whole concentration range is satisfying, especially considering the extremely low concentrations tested (average coefficient of variation %CV$_{av}$ = 25%, $n$ = 3).

Figure 4e shows bright-field (1) and fluorescence (2, 3) optical microscopy images of the cross-section of a LbL-biofunctionalized PSi interferometer after injection of 83 nM and 8.3 $\mu$M streptavidin labeled with sulfo-rhodamine. Control images are shown in Supplementary Figure 2. By comparison of fluorescence and bright-field images, it is apparent that binding of streptavidin uniformly occurs throughout the whole PSi thickness, both at the lower and the higher concentrations tested by fluorescence measurements. Negative control fluorescence images of bare (i.e., non-labeled) 8.3 $\mu$M streptavidin binding are shown in Supplementary Figure 2e,f.

A theoretical LoD is calculated from the calibration curve by extrapolation of the concentration corresponding to the IAW$_{LoD-IAW0}$ value for which SNR = $3.3\sigma_{IAW0}$ = 0.013 a.u., which gives rise to a minimum detectable concentration of $6 \times 10^{-4}$ nM (i.e., 600 fM). This is well consistent with the value obtained using the formula: LoD = $3.3\sigma_{IAW0}/S$ = $2.6 \times 10^{-4}$ nM (i.e., 260 fM), being $S$ = 49.16 nM$^{-1}$ the sensitivity of the PSi biosensor around a streptavidin concentration of 600 fM.

To the best of our knowledge, this is the lowest detected concentration using PSi as transducer in label-free and affinity biosensing since the first seminal Sailor group research paper, in which fM concentrations were detected exploiting charge-carrier mobilization induced by biomolecular complexation during the affinity bindings[42].

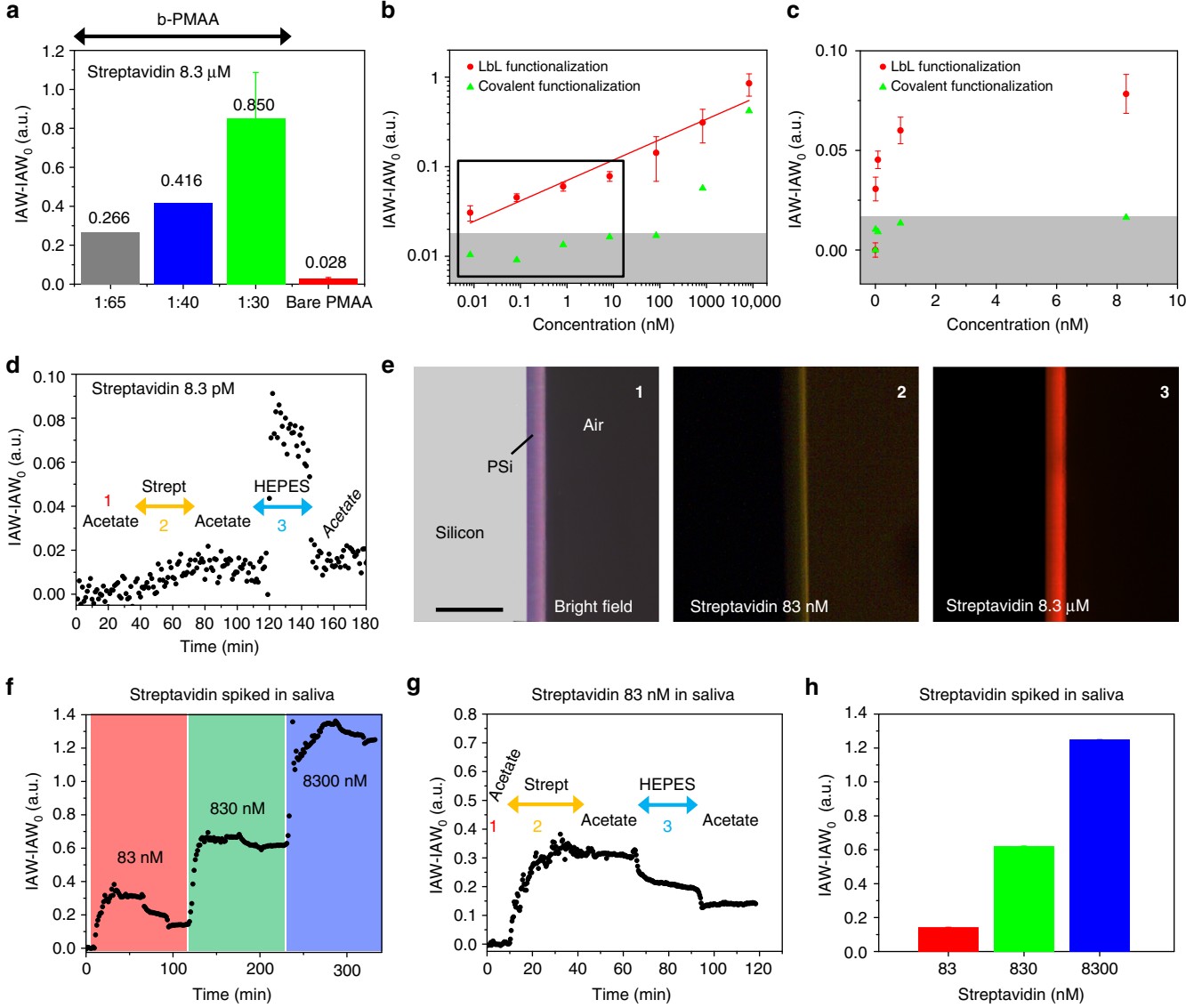

**Fig. 4** High-sensitivity biosensing with LbL-biofunctionalized PSi interferometers. **a** IAW-IAW$_0$ values recorded on LbL-biofunctionalized PSi interferometers upon affinity interaction with 8.3 µM streptavidin with b-PMAA at different biotinylation degrees, namely 1:65, 1:40, and 1:30. The IAW-IAW$_0$ value recorded on PSi interferometers LbL-coated with bare PAH/PMAA (non-biotinylated) due to unspecific adsorption of 8.3 µM streptavidin is also reported. **b** Calibration curve (IAW-IAW$_0$ vs. streptavidin concentration, log-log scale) of LbL-biofunctionalized PSi interferometers coated with b-PMAA (1:30) (red dots) for streptavidin concentration in the range 8.3 pM–8.3 µM; data are best-fitted with a power law (red solid trace). The calibration curve of PSi interferometers biofunctionalized using silane-based chemistry is also reported (green triangles). The gray region represents the 3.3$\sigma_{IAW0}$ noise level recorded in acetate buffer. **c** Magnification of the calibration curves in **b** for streptavidin concentration in the region 0–10 nM (linear scale). Data in **a**–**c** are provided as average values over three replicates with error bars representing one standard deviation. **d** Sensorgram (IAW-IAW$_0$ vs. time) acquired on LbL-biofunctionalized PSi interferometers coated with b-PMAA (1:30), showing specific detection of streptavidin 8.3 pM (i.e., the lowest tested concentration): signal stabilization in acetate buffer (step 1); injection of streptavidin and affinity interaction of streptavidin with b-PMAA (step 2); repulsive rinsing in HEPES buffer and removal of streptavidin unspecifically adsorbed on b-PMAA (step 3). **e** Bright-field (1) and fluorescence (2, 3) images of the cross-section of the oxidized PSi interferometer LbL-coated with PAH/b-PMAA after affinity binding of 83 nM (2) and 8.3 µM (3) sulfo-rhodamine-labeled streptavidin (scale bar is 15 µm). **f** Sensorgrams (IAW-IAW$_0$ vs. time), acquired on LbL-biofunctionalized PSi interferometers coated with b-PMAA upon consecutive injections of diluted saliva spiked with 83, 830, and 8300 nM of streptavidin. **g** Sensorgram (IAW-IAW$_0$ vs. time) of the injection of diluted saliva spiked with streptavidin 83 nM, highlighting the different injection steps. **h** Bar graph of IAW-IAW$_0$ signals acquired after injection of diluted saliva spiked with streptavidin 83, 830, and 8300 nM. Data are provided as average values of IAW-IAW$_0$ fluctuations over time (data points = 30) with error bars representing one standard deviation

To fairly assess advantages of the LbL biofunctionalization strategy for biosensing applications with respect to covalent functionalization chemistry (i.e., organosilane chemistry), we prepared a biotinylated PSi interferometer (as control) using (3-aminopropyl)triethoxysilane (APTES) as linker between the oxidized surface of PSi and biotin 3-sulfo-*N*-hydroxysuccinimide ester sodium salt. Effectiveness of covalent biofunctionalization was verified step-by-step using EOT reflectance spectroscopy (Supplementary Figure 5). Biotinylated control interferometers were then tested for streptavidin detection in the range of concentration 8.3 pM to 8.3 µM, using IAWRS.

Figure 4b (green triangles) shows the calibration curve achieved for the PSi interferometer prepared using covalent chemistry, superposed to that achieved using LbL biofunctionalization. In this case, only streptavidin concentrations >83 nM are above the noise region (gray area in Fig. 4b), so that they are reliably detected. These concentrations are at least five orders of magnitude higher than those measured using LbL-biofunctionalized interferometers. Notice that, the noise level of the $IAW_0$ value measured for the control interferometer is $3.3\sigma_{IAW0} = 0.017$ a.u., which is statistically equal to that obtained for LbL-biofunctionalized interferometers, i.e., 0.013 a.u.

These experimental results confirm that LbL biofunctionalization produces a significant improvement of the analytical performance of optical label-free PSi biosensors, in terms of sensitivity and LoD, at least, compared with that achieved using the commonly employed covalent silane chemistry.

**Assessment of LbL-biofunctionalized PSi interferometers in saliva.** Stability and effectiveness of LbL biofunctionalization is eventually tested in a real and complex human matrix, namely non-filtered saliva. Saliva is an exocrine and oral fluid secretion that is attracting increasing attention in clinical chemistry and forensic toxicology being a well-established and non-invasive alternative to serum and plasma[43].

The chemical composition of saliva consists of 99.5% water and 0.5% solid components, roughly. The solid components consist, in turn, of 0.1% electrolytes (i.e., calcium, magnesium, sodium, potassium, chloride, bicarbonate, phosphate), 0.3% proteins (i.e., enzymes, IgG, IgA, antimicrobial factors, mucosal glycoproteins, albumin, polypeptides, oligopeptides), and for the remaining 0.1% of small organic molecules derived from metabolic activity (i.e., glucose, citrate, lactate, ammonia, urea, uric acid creatinine cholesterol and cyclic adenosine monophosphate)[44]. In addition, saliva also contains buccal epithelial cells and blood leukocytes[45].

Saliva samples (1-2 mL) collected from a volunteer were immediately vacuumed, then diluted 1:10 in acetate buffer (i.e., the running/binding buffer of this work). Dilution has the twofold aim of: (1) reducing the concentration of solid components, namely proteins and small organic molecules, from 0.4% (i.e., 4000 μg mL$^{-1}$) to 0.04% (i.e., 400 μg mL$^{-1}$), to yield it comparable to the highest concentration of biomolecules tested in acetate buffer, namely 500 μg mL$^{-1}$ of streptavidin, pepsin, and BSA; (2) decreasing saliva viscosity, to reduce possible fluidics (e.g., microchannel clogging) and optical (e.g., unwanted refractive index changes) effects induced by a highly concentrated solid compound and cells. The diluted saliva (without any further filtration or purification) is either directly used for evaluation of the stability of LbL-biofunctionalized PSi interferometers or spiked with streptavidin at concentrations of 83, 830, and 8300 nM to evaluate its selective detection in a complex body fluid.

Stability of the LbL nano-assembly in saliva is investigated in terms of changes in the EOT value in acetate buffer of LbL-coated PSi interferometers upon injection of saliva at 5 μL min$^{-1}$ for 40 min, using the same injection protocol described for BSA and pepsin control experiments (see section Optimizing specific/unspecific binding with repulsive rinsing). The LbL-biofunctionalized PSi interferometers were secured into a flow cell and stability was investigated through FFT reflectance spectroscopy, which allows refractive index variation due to deterioration of the LbL assembly to be reliably detected (Supplementary Figure 6a). $EOT-EOT_0$ values recorded before (i.e., $0.08 \pm 2.98$ nm) and after (i.e., $-4.1 \pm 5.6$ nm) injection of saliva (over three replicates, in acetate buffer) confirm a good stability of the LbL nano-assembly. Indeed, the EOT changes measured in saliva are not statistically different (Student's $t$-test

confidence level = 99%) from that recorded injecting only acetate buffer over 80 min ($EOT-EOT_0 = -3.8 \pm 12.6$ nm, Fig. 3b).

To better quantify unspecific adsorption of solid components of saliva on the LbL-biofunctionalized PSi interferometers, IAWRS was used to analyze the reflectance spectra upon saliva injection over time, as it allows a higher sensitivity with respect to FFT reflectance spectroscopy to be achieved[40]. Supplementary Fig. 6b shows the typical sensorgram (i.e., $IAW-IAW_0$ vs. time) recorded for saliva using IAWRS, from which it is apparent that biomolecules (present in saliva) unspecifically adsorbed on the LbL coating after saliva injection are efficiently removed through the repulsive rinsing step in HEPES buffer. Remarkably, the residual IAW value (in acetate buffer) after saliva injection (i.e., $IAW-IAW_0 = 0.090 \pm 0.013$ a.u.) is smaller than that achieved with BSA and only three times that measured for pepsin, at about same concentration. This is an important result, especially considering both complexity and heterogeneity of saliva, which contains about 400 μg mL$^{-1}$ (after 1:10 dilution) of biomolecules and small organic molecules derived from metabolic activity, in addition to cells. Notice that, the use of diluted saliva does not produce significant oscillations of the IAW signal in acetate buffer after the rinsing step (i.e., noise level after saliva injection and rinsing $\sigma_{IAW} = 0.004$ a.u., which is the same value achieved injecting only acetate buffer, namely $\sigma_{IAW0}$).

Analytical performance of the LbL-biofunctionalized PSi interferometers in saliva spiked with 83, 830, and 8300 nM of streptavidin are then evaluated using the same protocol described above. Figure 4f shows the sensorgram recorded for three consecutive injections of saliva spiked with 83, 830, and 8300 nM streptavidin, which clearly highlights that all the tested streptavidin concentrations are well discriminated in saliva. Remarkably, the IAW signal monotonically increases as the streptavidin concentration in saliva increases, reaching IAW values (measured after the repulsive rinsing in HEPES) comparable to those achieved in acetate buffer at the same streptavidin concentrations (see Fig. 4b). $IAW-IAW_0$ values recorded for 83, 830, and 8300 nM streptavidin in saliva using LbL-biofunctionalized PSi interferometers are summarized in Fig. 4h. A detailed view of the sensorgram recorded for saliva spiked with 83 nM streptavidin is shown in Fig. 4g. Conversely to what happens in blank saliva (with no streptavidin), during the injection of saliva spiked with streptavidin the IAW value shows noticeable oscillations and the adsorption kinetics results to be faster than in blank saliva. Nonetheless, the signal-to-noise ratio of the IAW signal measured in acetate buffer after injection of either blank or streptavidin-spiked saliva remains unchanged. We argue that, in the presence of streptavidin, competitive processes occur at the LbL nano-assembly between faster specific binding of streptavidin with biotin and slower unspecific binding of interfering biomolecules, present in saliva, with PMAA. This is consistent with the reduction of both IAW oscillations and IAW unspecific signal as the streptavidin concentration in saliva increases.

## Discussion
In this work, LbL biofunctionalization of nanostructured materials by self-assembling of oppositely charged polyelectrolytes engineered with suitable bioreceptors is successfully demonstrated as a robust and effective alternative route to covalent chemistry towards the development of optical label-free affinity biosensors with improved performance, in terms of sensitivity and selectivity. In particular, high sensitivity is enabled through self-regulated deposition of a conformal and stable nano-assembly featuring a prescribed density of bioreceptors; on the other hand, a repulsive rinsing step removing most of the

biomolecules unspecifically bound to the nano-assembly allows a high selectivity to be achieved.

The proof-of-concept demonstration is given on a nanostructured PSi interferometer, which was employed for the development of a label-free biotin–streptavidin affinity bionsensor. LbL biofunctinalization is shown to improve the performance of PSi interferometers of a factor $10^5$ with respect to the covalently biofunctionalized counterpart used as control, both in terms of LoD (600 fM) and sensitivity (49 nM$^{-1}$). This also represents a 300-fold improvement with respect to the best PSi biosensors reported in the current literature, pushing PSi biosensors to performance comparable to that of best state-of-the-art nanostructured photonic and plasmonic platforms for biosensing, namely, SPR, LSPR, interferometers, ring resonators, optical fibers, and photonic crystals.

Notice that, these latter platforms commonly require accurate surface patterning and complex interrogation setups, which is not the case of PSi interferometers. Further, both stability and analytical performance of LbL-biofunctionalized PSi interferometers are successfully assessed in a complex body fluid, namely saliva (non-filtered) spiked with streptavidin. This makes LbL-biofunctionalized PSi interferometers a very appealing platform for low-cost and high-sensitivity point-of-care biosensing.

By building on these results, it is straightforward to envisage application of the LbL biofunctionalization route to other state-of-the-art photonic and plasmonic platforms for biosensing, both bulk and nanostructured, which would possibly lead to a significant improvement of their performances in label-free affinity biosensing, beyond those achieved using standard covalent chemistry. On the other hand, further development of the LbL biofunctionalization route by eitherengineering polyelectrolytes with different bioreceptors, e.g., DNA strands, or using different/stronger polyelectrolytes, e.g., polystyrene sulfonate (pKa = 1), can be foreseen, which would significantly broaden flexibility and applications of LbL biofunctionalization.

## Methods

**Materials and chemicals**. Silicon wafer boron doped, <100> oriented and with resistivity of 0.8–1.2 m$\Omega$-cm (p++-type), were purchased from Siltronix Silicon Technologies (France). Aqueous hydrofluoric acid (HF, 48%) was purchased from Merck Millipore (Massachusetts, USA). Absolute ethanol (99.9%), isopropyl alcohol (99.5%), and diethylether were purchased from Carlo Erba Reagents S.r.l (Italy). Streptavidin from Streptomyces avidinii (pI = 5.0, MW = 60 kDa) to be used for affinity biosensing and to be linked with Sulfo-rhodamine was purchased from Fisher Scientific. Toluene (99.8%), APTES (99%), sodium hydroxide (NaOH > 98%), sodium chloride (NaCl 99%), acetic acid (CH$_3$COOH, 99.5%), sodium acetate (CH$_3$COONa 99%), 4-(2-hydroxyethyl)-1-piperazineethanesulfonic acid (HEPES 99.5%), biotin 3-sulfo-N-hydroxysuccinimide ester sodium salt, BSA (pI = 4.7, MW = 66 kDa), pepsin (pI = 1, MW = 35 kDa), biotin amine, diethylether (Et$_2$O), trifluoroacetic acid (TFA), and 2-(4-hydroxyphenylazo)benzoic acid (HABA) were purchased from Sigma Aldrich (Germany). PAH (40,000 Da) was purchased from Beckmann-Kenko (Germany). PMAA was purchased from Polysciences (Germany). N-hydroxysuccinimide (NHS), N,N'-dicyclohexylcarbodiimide (DCC), N-boc-ethylenediamine, N-(3-dimethylaminopropyl)-N'-ethylcarbodiimide hydrochloride (EDC), and N,N-dimethylformamide (DMF) were purchased from Fluka. The labeling procedures were performed by common coupling procedures[46], with subsequent dialysis using membranes of 15 kD MWCO (Aldrich). Acetate buffer (10.0 mM CH$_3$COOH/CH$_3$COONa) adjusted to a pH = 5 was used for stock solution preparation from lyophilized streptavidin (20 mg mL$^{-1}$) and stored at −20 °C. Acetate buffer (10.0 mM CH$_3$COOH/CH$_3$COONa) with the addition of 100 mM NaCl was employed as runner/binding buffer during biosensing measurements and for the serial dilution of the streptavidin stock solution. Sodium acetate buffer (50 mM CH$_3$COONa) with the addition of 0.2 M NaCl and adjusted to a pH = 5.6 was used for the solubilization of PAH, PMAA, and PMAA biotinylated polymers (1 mg mL$^{-1}$) for LbL coating. HEPES buffer (10.0 mM) with 100 mM NaCl and adjusted to a pH = 7.4 was used for the rinsing step after the affinity interaction between biotin and streptavidin.

PBS buffer (100 mM) with 100 mM NaCl and adjusted to a pH = 7.4 was used for the stability test on the LbL assembly and for the biotinylation of the silanized PSi oxidized surface.

All buffers were prepared in deionized water (DIW), filtered using syringe filters (Minisart® NML Syringe Filters 1.20 μm) and pH-adjusted with NaOH (5 M) and HCl (1 M) aqueous solution.

**Preparation and oxidation of PSi interferometers**. PSi samples were prepared by anodic etching of polished silicon wafer (1.5 × 1.5 cm) with a solution of HF (48%): EtOH, 3:1 v/v at 18 °C. A two-electrodes Teflon cell equipped with a platinum wire cathode and an aluminum flat anode was employed for the etching of the silicon samples over a circular area of 0.567 cm$^2$. A Keithley 2602A SourceMeter was used to set the etching current density and measure the etching voltage.

A first PSi sacrificial layer was etched at 700 mA cm$^{-2}$ for 10 s and dissolved by alkaline dissolution with a solution of NaOH (1 M):EtOH 9:1 v/v to avoid the formation of a top parasitic layer (with pores of a few nanometers in diameter) restricting the diffusion of both LbL polymers, e.g., PAH and PMAA, and target biomolecules, e.g., streptavidin (~5 nm diameter), inside the PSi layer underneath[47]. The silicon samples were rinsed with abundant DIW and ethanol, and dried under a gentle nitrogen flow.

The PSi sensing layer (i.e., the PSi interferometer) was then etched at 600 mA cm$^{-2}$ for 25 s on the so-processed silicon samples, rinsed with isopropanol and diethylether, and gently dried under a nitrogen flow to achieve a crack-free PSi layer. Eventually, a thermal oxidation of the PSi interferometer was carried out in a muffle furnace (ZB 1, ASAL, Italy) at 750 °C for 1 h (ramp-up/ramp-down 12 °C min$^{-1}$) in room atmosphere.

**Characterization of PSi interferometers**. Reflectance spectra of the PSi interferometers were acquired in air (both before and after oxidation) in the wavelength range [400−1000 nm] using an optical setup consisting of a UV−VIS spectrometer (USB2000-VIS-NIR-ES), a bifurcated fiber-optic probe (QR200−7-VIS-BX), and a halogen lamp source (HL-2000) purchased from Ocean Optics (USA). Light exiting the halogen lamp source is fed orthogonally onto the PSi surface through one arm of the fiber-optic probe; the light reflected from the PSi layer is collected into a UV−VIS spectrometer through the other arm of the fiber-optic probe. Acquisition parameters for reflection spectra were: integration time 2 ms, average scan number 5, boxcar width 5, with the spectrometer working in photon counts mode.

Porosity of as-prepared PSi interferometers was estimated by best-fitting of the reflectance spectra of PSi layers acquired before oxidation, using a home-made software (Matlab, MathWorks, USA)[48]. Morphological characterization of as-prepared PSi interferometers (i.e., diameter and thickness of the pores) was performed on PSi layers before oxidation, using a scanning electron microscope (SEM, JSM-6390, JEOL, Italy) at an accelerating voltage of 5 kV and an Atomic Force Microscope (AFM Edge, Brucker, USA) in tapping mode. Distribution of the pore diameters was obtained from the analysis of both SEM and AFM images with ImageJ[49].

Reproducibility of PSi preparation and oxidation was assessed by FFT of the reflectance spectra of PSi interferometers, acquired before and after oxidation, through calculation of EOT values (see section FFT reflectance spectroscopy).

**Biotinylation of PMAA**. Biotinylation of PMAA was carried out under dry conditions. In all, 1.22 g biotin (5.0 mmol, 1.00 eq) was added to a two-neck round-bottom flask, followed by 0.60 g NHS (5.2 mmol, 1.04 eq) dissolved in 20 mL DMF. In total, 1.20 g DCC (5.8 mmol, 1.16 eq) in 8 mL DMF was added dropwise using a dropping funnel, which was then rinsed with 2 mL DMF to transfer all the material. The mixture was heated at 75 °C until consumption of biotin checked by thin layer chromatography (TLC) (acetonitrile, water, acetic acid, 90:10:1). After 1 h, 0.80 g N-boc-ethylenediamine (792 μL, 5.0 mmol, 1.00 eq) in 4 mL DMF was added subsequently to the suspension, followed by 2.8 mL triethylamine (20.0 mmol, 4.00 eq). The reaction mixture was heated at 75 °C until consumption of the intermediate checked by TLC (acetonitrile, water, acetic acid, 90:10:1). After 30 min, the reaction mixture was filtered to remove the dicyclohexylurea by-product and the filtrate was concentrated under reduced pressure and an off-white solid was obtained. The crude product was re-crystallized from isopropanol. The pale-yellow solid was washed with Et$_2$O to finally obtain a white solid. To the white solid 40 mL DCM and 12 mL TFA (156.8 mmol, 31.20 eq) were added and the mixture was refluxed for 10 min. The solvent was evaporated under reduced pressure and the product was freeze-dried to obtain the biotin-amine TFA salt as an off-white solid in 50 % yield (0.96 mg, 2.5 mmol).

Biotinylated PMAA polymers with theoretical labeling degrees of 1:20, 1:10, and 1:5 (biotin:PMAA monomer) were synthesized by amide coupling of PMAA with different amounts of biotin-amine TFA salt. For each labeling degree, 50 mg PMAA were dissolved in 0.5 mL 0.1 M HEPES pH 7.4. To each sample, 60 mg EDC (0.31 mmol) in 250 μL 0.1 M HEPES pH 7.4 and 17.5 mg NHS (0.15 mmol) in 250 μL 0.1 M HEPES pH 7.4 were added. The mixture was stirred for 1 h and an additional 1 mL 0.1 M HEPES pH 7.4 was added. Biotin-amine TFA salt (10 mg, 25 mg, and 50 mg) dissolved in 1 mL DMSO was added in order to obtain three different polymers with three different biotinylation degrees. The mixtures were stirred for 2 days at room temperature and dialyzed for 1 week using a dialysis membrane and concentrated under reduced pressure to obtain the biotinylated PMAA polymers.

**Labeling degree of biotinylated PMAA**. The experimental degree of biotinylation was determined using the 4′-hydroxyazobenzene-2-carboxylic acid (HABA) assay. The HABA assay is based on the binding of HABA to streptavidin, which generates a colored complex absorbing at 500 nm. Biotin has the ability to displace HABA from the complex, proportionately decreasing the adsorption[50]. The experimental degrees of biotinylation were 1:65, 1:40, and 1:30.

A solution containing 100 μL of streptavidin 0.5 mg mL$^{-1}$ in 50 mM K$_3$PO$_4$ and 150 mM NaCl at pH 6 was prepared and 2.5 μL of HABA 10 mM in 10 mM NaOH were added. To this solution 1 μL of biotinylated PMAA 10 mg mL$^{-1}$ in water was added. The absorbance at 500 nm was measured and the amount of biotin was calculated using a calibration curve.

**LbL functionalization of PSi interferometers**. Oxidized PSi interferometers were first LbL coated with PAH (1 mg mL$^{-1}$), which is a positively charged polyelectrolyte, exploiting electrostatic attraction with the negatively charged oxidized surface of PSi interfeomters. The PAH solution (100 μL) was drop-cast onto the oxidized PSi interferometers and left incubating for 2 h at 18 °C to ensure full infiltration of the nanopores of the PSi layer. The samples were then abundantly rinsed with DIW and ethanol, and gently dried under a nitrogen flow. The PAH-coated PSi interferometers were then LbL coated with either bare and biotinylated PMAA at 18 °C, which is a negatively charged polyelectrolyte, exploiting electrostatic interaction with PAH. The PMAA solution (100 μL, 1 mg mL$^{-1}$) was dropped onto the PAH-PSi interferometers and left incubating for 2 h at room temperature to ensure full infiltration of the PAH-coated nanopores. The samples were then abundantly rinsed with DIW and ethanol, and gently dried under a nitrogen flow. The same coating protocol was followed also for sulfo-rhodamine-labeled PAH and fluorescein-labeled PMAA charged polymers.

Reflection spectra of PSi interferometers after each LbL-coating step (i.e., PAH and PMAA) were acquired in air (using the same protocol reported in section *Characterization of PSi interferometers*). Successful LbL coating of PSi interferometer was confirmed by FFT of the acquired reflectance spectra, through calculation of the EOT value (see section *FFT reflectance spectroscopy*). Optical microscope (Leica DM2500 M), equipped with a filter cube for green and red fluorescence, was used to further corroborate LbL coating of PSi interferometers, using sulfo-rhodamine-labeled PAH and fluorescein-labeled PMAA.

**Covalent biotinylation of oxidized PSi interferometers**. The covalent biotinylation of the oxidized PSi surface was based on the silanization chemistry with APTES (silane).

Oxidized PSi interferometers were incubated with 1% APTES (10 mL) in anhydrous toluene for 1 h[51]. After incubation, the PSi samples were abundantly rinsed with toluene and ethanol, and gently dry under a nitrogen flow. PSi samples were baked at 150 °C for 30 min to further stabilize the APTES layer[52], then rinsed in PBS buffer to remove either hydrolyzed or multilayered silanes. FFT reflectance spectroscopy was employed to confirm the occurred silanization of PSi inteferometers (Supplementary Figure 5a, b).

Covalent biotinylation of APTES-coated PSi interferometers was carried out using Biotin 3-sulfo-*N*-hydroxysuccinimide ester sodium salt solubilized in PBS (10 g mL$^{-1}$ ~ 22.5 mM). PBS buffer (pH = 7.4) was chosen taking into account that Biotin 3-sulfo-*N*-hydroxysuccinimide couples to primary amines in the pH range 6.5–8.5. The covalent biotinylation of the amino-group was confirmed by IAWRS (Supplementary Figure 5c).

**FFT reflectance spectroscopy**. FFT of the reflectance spectra of PSi interferometers was performed to calculate the EOT values, namely, 2nL, where $n$ = effective refractive index and $L$ = thickness of the PSi layer, using a home-made software (MatLab, MathWorks, USA). The wavelength axis of the reflectance spectrum was first inverted ($x$ axis changed from wavelength to 1/wavelength) to obtain a wavenumber axis. A cubic-spline interpolation of reflectance data was then carried out to obtain a dataset (reflection, wavenumber) spaced evenly (sample-to-sample distance $8.57 \times 10^{-7}$ nm$^{-1}$). A Hanning window was applied to the reflectance spectrum, which was zero padded to $2^{24}$. Eventually, application of the FFT algorithm to the zero-padded reflectance spectrum yielded the Fourier transform amplitude and phase ($y$ axis in the Fourier transform domain) as a function of 1/wavenumber ($x$ axis in the Fourier transform domain), with spatial resolution of about 0.07 nm. The EOT value is obtained as the value of the 1/wavenumber axis ($x$ axis) in the Fourier transform domain for which the main peak in the Fourier transform amplitude ($y$ axis) occurs.

**Biosensing protocol of biofunctionalized PSi interferometers**. The fiber-optic setup previously described in section *Characterization of PSi interferometers* was integrated with a flow cell system (as depicted in the Supplementary Information of[53]) for the monitoring of: (1) the unspecific interaction between PMAA and proteins (i.e., streptavidin, pepsin and BSA, at a concentration of 500 μg mL$^{-1}$); (2) the selective interaction between b-PMAA (1:65), b-PMAA (1:40), and b-PMAA (1:30) and 8.3 μM (500 μg mL$^{-1}$) streptavidin; (3) the biosensing interaction between streptavidin, at concentration ranging from 8.3 pM to 8.3 μM, and b-PMAA (1:30).

The PSi interferometers were secured into a home-made Plexiglas flow cell provided with a PDMS o-ring (volume of 100 μL). Solutions under test were

injected in the flow cell through a syringe pump (Nexus 3000, Chemyx Inc., USA) working in withdraw mode.

First, acetate buffer (running buffer) was injected at a flow rate of 5 μL min$^{-1}$ for a warm-up time of 60 min, to make sure that both fluidic and thermal transients were over. After warm-up, acetate buffer was further injected for 30–40 min, then protein solutions (i.e., streptavidin, BSA, or pepsin) at different concentrations were injected for 40 min at a flow rate of 5 μL min$^{-1}$ (200 μL total volume injected). After injection of a protein solution with a specific concentration, a first rinsing step in acetate buffer was carried out at a flow rate of 100 μL min$^{-1}$ for 20–30 min to remove both weakly adsorbed proteins on the PMAA matrix and unbounded proteins present in the bulk media. A further repulsive rinsing step with HEPES buffer was then performed at a flow rate of 100 μL min$^{-1}$ for 20–30 min, in order to desorb proteins nonspecifically adsorbed on PMAA (either biotinylated or bare). Eventually, acetate buffer was injected again in the flow cell at a flow rate of 5 μL min$^{-1}$ for 20 min, before injection of a protein solution with a novel concentration.

Notice that, the repulsive rinse with HEPES buffer, which was specifically developed for the LbL assembly, was bypassed for covalent-biotinylated PSi interferometers.

Reflection spectra of PSi interferometers were acquired every minute over the whole biosensing experiment. Acquisition parameters of reflection spectra were: integration time 2 ms, average scan number 5, boxcar width 5, with the spectrometer working in normalized reflection mode.

**Saliva sampling and biosensing measurements in saliva**. Saliva (1–2 mL) was sampled from a health human male volunteer (after tooth brushing in the morning, 1 h before sampling), upon written informed consent. The saliva was spit in a sterilized Petri dish and then immediately vacuumed for 30 min to remove air bubbles that could compromise the correct operation mode of the fluidic systems and the correct interference spectra acquisition. An aliquot of the sampled saliva (100 μL) was diluted 1:10 in acetate buffer (running/binding buffer reported in Materials and chemicals paragraph). An aliquot of the diluted real matrix (i.e., 200 μL) was directly injected (without any filtration or purification) in the flow cell for testing the LbL assembly stability and using the same injection protocol (with repulsive rinse) described in Biosensing with LbL-biofunctionalized PSi interferometers paragraph. The stability of the LbL assembly was evaluated monitoring the EOT changes of the LbL-coated PSi surface according to Stability of LbL assembly with pH and ionic strength paragraph reported in *Results* section.

Once the stability was assessed, the binding properties of the biotinylated surface in complex matrix were evaluated testing aliquots of saliva (1:10 diluted in acetate buffer) spiked with 83, 830, and 8300 nM (i.e., 5, 50, and 500 μg mL$^{-1}$) of streptavidin accordingly to the injection protocol described in Biosensing with LbL-biofunctionalized PSi interferometers paragraph. The bindings curves were monitored by IAW reflectance spectroscopy and the obtained IAW-IAW$_0$ signals were compared with the same obtained in only acetate buffer for the same streptavidin concentrations.

**Biosensing measurement by IAWRS**. The IAW Reflectance Spectroscopy[33,41] was used to monitor: stability of the LbL assembly at different pH and ionic strength buffers; unspecific interaction between streptavidin, pepsin, BSA, and PMAA; biosensing measurements with streptavidin on LbL-biofunctionalized PSi interfermeters; as well as, biosensing measurements with streptavidin on covalently biotinylated PSi interferometers.

Reflectance spectra acquired on PSi interferometers were normalized with respect to a reference mirror (Ø1"Protected Silver Mirror, Thorlabs, USA). Spectral interferograms were calculated by subtraction (intensity, wavelength by wavelength) of the reflectance spectrum acquired after each injection (i.e., protein, buffer, linkers) from the reference reflectance spectrum acquired in the running buffer (i.e., acetate). An interferogram was also calculated for the running buffer (namely, blank interferogram) by subtraction of the reflectance spectrum acquired in acetate buffer before starting injecting the protein solutions ($t = 100$ min) from the reflectance spectrum acquired after the 60-min-long warm-up time ($t = 60$ min). Each interferogram was then corrected for its average value, which was subtracted from the interferogram to limit artifacts due to spectral reflection changes of intensity induced by biomolecules adsorption on the top of the PSi sensing layer, and/or on the quartz window of the flow cell. Eventually, the output signal, namely, IAW, was obtained by application of the absolute value function to each so-processed interferogram and subsequent calculation of the average value of the resulting interferogram over the spectral range of interest [500−800 nm]. A schematic description of the whole IAWRS strategy is shown in Supplementary Figure 7, with specific reference to an LbL-biofunctionalized PSi interferometer with b-PMAA (1:30) upon injection of the highest steptravidin concentration tested (i.e., 8.3 μM).

**Study approval**. Saliva was obtained from a volunteer upon written informed consent prior to inclusion in the study. The study protocol was approved by the Committee on Bioethics of the University of Pisa (review no. 3/2018).

## Data availability
The authors declare that all relevant data supporting the findings of this study are available within the paper and its Supplementary Information files.

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

## Acknowledgements

This work was jointly supported by the EU H2020 ETN SYNCHRONICS under grant agreement 643238 and by the Italian Minister for University and Research (MIUR) Futuro in Ricerca (FIR) programme under grant number RBFR122KL1 (SENS4BIO). We thank Dr. F. F. R. Toia (STMicroelectronics, Milan) for technical assistance with SEM images.

## Author contributions

G.B. and S.M. contributed to the design of the study. S.M. carried out the experiments and collected the data, which were analyzed and interpreted in conjunction with G.B., L. M.S., and V.R. S.M. and G.B. wrote the manuscript and all authors were involved in revisions. G.E. and A.D. prepared, characterized, and engineered the polyelectrolytes, under L.D. supervision. G.B. supervised the research work.

## Additional information

**Competing interests:** The authors declare no competing interests.

