## [Peer Review File · Nature Communications]

Reviewers' comments:

Reviewer #1 (Remarks to the Author):

The authors present an interesting work on the development of biosensors based on nanostructured materials.

In this sense, there are other techniques that enable higher sensitivities in the order of sub-attomolar for streptavidin-biotin interactions so I would encourage the authors to modify the title accordingly and introduce PSi in the title. Could the authors mention and compare some of these techniques and results more in detail?

The main novelty of the paper relies on the utilization of LbL electrostatic nano assembly technique for the fabrication of the proposed device. Neither the utilization of the sensing structure nor the LbL technique are new and they have been described before in many papers from different authors. However, this paper has some novelty on the approach presented for the utilization of the LbL technique taking into account the specific orientation issues of these devices as well as the complex chemistry associated to PSi structures that commonly rely on covalent bondings. Concerning previous comments. Covalent bonding has been traditionally described as stronger, with definite shape and preferred for many applications to ionic bondings. Do the authors claim in this paper that electrostatic bonding is preferable to covalent bonding? Could the authors clarify this? Could the authors clearly identify some of the drawbacks of the utilization of this technique (I suppose there are some)?

The authors present a simple case on biotin-streptavidin affinity of the device showing very interesting and promising results. However, as the authors indicate in the manuscript, streptavidin-biotin bond is highly stable and it would be interesting if the authors could test some other bindings, such as IgG (used commonly as a gold standard in different sensing platforms) or comment the major drawbacks to perform other affinity assays using current approach, such as obtaining the adequate polymer or the right orientation for a correct binding.

In addition there are some questions that need to be addressed prior to publication:

1.- It would be interesting if the authors could describe schematically the structure formed onto the PSi substrate. Does it consist of a single PAH and B-PMAA layer? Does it have several bilayers?

2.- Could you please indicate the parameters for the fabrication of the structures (pH of the solutions, concentrations, temperature, etc.).

3.- Could you please comment on the effects of the variation of the fabrication parameters on the final properties of the structure and the implications on the sensing performance of the final device

4.- Could you please comment on the importance of the utilization of HEPES buffer after acetate buffer? Could you please discuss the chemical background of the repulsive behaviour of HEPES buffer? Is it a common procedure?

5.- PAH and PAA have been shown as pH sensitive polymers. Could you please indicate the problems associated to this behaviour on the sensing performance of the device?

6.- PAH and PAA are weak polyelectrolytes and their structure can easily change during fabrication if pH is not adjusted properly giving rise to undesired structural properties. How do the authors deal with this problem?

Following the previous comment. Would it be possible to replace PAH and PAA by other strong

polyelectrolytes, such as PEI and PSS in order to obtain a more robust device and more repetitive results?

7.- On Figure S4b) there is a signal shift around minute 70. Could you please comment?

Reviewer #2 (Remarks to the Author):

The authors show a proof-of-concept of the advantages of LbL biofunctionalization, by using nanostructured PSi interferometer, which is "biofunctionalized" for the affinity detection of streptavidin via LbL nano-assembling of PAH with b-PMAA. The interferometers results to be very effective and highly robust, with a detection limit of 600 fM.

It is an advance in the field of diagnose. The claims are novel and they could be of interest to others in the community and the wider field, including chemical biology, medicine and biosensor chemistry.

However, please consider the following points:

1 - LbL technique is very useful to create interfaces for diagnoses. In the introduction part the authors should provide a more deeply and concise information in how does LbL can solve the problem in high-specificity detection of streptavidin. In the context of nanomaterials for diagnosis, what is the challenges and applications in smart devices based on molecular recognition?

2- After LBL approach, the stability of the film must be evaluated in a real biological media, for example, blood serum, cellular fluid, among other biological fluids.

3 – About interference adsorption, could the authors provide a quantitative data?

4 – Statistic data should present SD, error and it can be used in the 99% of confidence level in terms of t-Student confidence level. A bunch of data in this regard can give us a more realistic information. Please, consider appropriateness and validity of deeply statistical analysis, as well the high level of detail must be provided.

5 – I wonder to see if it is possible make the miniaturization of the Psi. How small it is possible? Can the authors show us some experiment and result about detector and device, with good stability and reproducibility in micro or sub-micro scale? Please comment it and if it is possible, show some results.

6 – This work is convincing, however, further "real" sample, with no time-consuming techniques to prepare samples would be required to strengthen the conclusions.

7 - It is true that "bioreceptors can be incorporated in individual films and that difficult to control (unreliable) chemistry steps, e.g. coupling of bioreceptors, can be determined in solution, before going to nanostructured surface." How does it can be compared with other systems quantitatively? The authors can sustain this information based on real data?

8 – The title: "Layer-by-Layer Nano-Assembly as a New Biofunctionalization Route For High-Sensitivity and High-Specificity Label-Free Affinity Biosensing with Nanostructured Materials" sounds very general if we compare it with conclusion. It can be fixed. Also, the "new" should be avoided.

Answers to Reviewer's comments

Reviewer #1

Comment: The authors present an interesting work on the development of biosensors based on nanostructured materials.

Answer: We would like to sincerely thank the reviewer for her/his positive feedback on our work.

Comment: In this sense, there are other techniques that enable higher sensitivities in the order of sub-atto-molar for streptavidin-biotin interactions so I would encourage the authors to modify the title accordingly and introduce PSi in the title. Could the authors mention and compare some of these techniques and results more in detail?

Answer: Following the reviewer's suggestion, the title has been changed to take into account the fact that we used nanostructured porous silicon to validate the proposed Layer-by-Layer (LbL) biofunctionalization route.

The title now reads: "Layer-by-Layer Nano-Assembly as a Biofunctionalization Route For High-Sensitivity and High-Specificity Label-Free Affinity Biosensing with Nanostructured Materials: The Case of Nanostructured Porous Silicon"

We fully agree with the reviewer that there are other approaches/techniques that enable higher sensitivity biosensing, with limit of detection down to the attomolar and sub-attomolar level, such as those based on gold nanoparticles signal amplification strategies [1] and ELISA test [2]. Nonetheless, it is important to remark that these techniques are not label-free (as for the biosensor of this work), as they make use of either fluorescent or non-fluorescent amplification labels to push sensitivity up and reduce, in turn, detection limit.

In fact, both fluorescent and tagged techniques have enabled great performance to be achieved in biosensing and several systems based on these approaches have been successfully commercialized over the last decades (e.g. ELISA). On the other hand, the use of tags in biomolecules requires time-consuming and money-consuming chemical modifications of biomolecules (either bioreceptor or target) with signal enhancer, which also increase the steric hindrance of the total assay and could play a major role in nanostructured materials.

Summarizing, although we fully agree with the reviewer on the existence of high sensitivity techniques for biosensing, these mostly rely on the use of labels for signal amplification and we do not feel like that a discussion of these techniques would fit well in this manuscript, as it appears to be out of the scope of this work, mainly focusing on the demonstration of LbL as a biofunctionalization approach for label-free biosensors.

On the other hand, main techniques and platforms for label-free biosensing (i.e. without labels) were taken into account, and their performance discussed in the final part of the introduction section, both for solid and nanostructured materials. As a matter of fact, Figure 1 summarizes performance of the main solid and nanostructured label-free optical platforms

using limit of detection (i.e. the ratio between 3 times the standard deviation of noise and the sensitivity of the sensor) as a figure of merit.

Comment: The main novelty of the paper relies on the utilization of LbL electrostatic nano assembly technique for the fabrication of the proposed device. Neither the utilization of the sensing structure nor the LbL technique are new and they have been described before in many papers from different authors. However, this paper has some novelty on the approach presented for the utilization of the LbL technique taking into account the specific orientation issues of these devices as well as the complex chemistry associated to P*Si* structures that commonly rely on covalent bondings. Concerning previous comments. Covalent bonding has been traditionally described as stronger, with definite shape and preferred for many applications to ionic bondings. Do the authors claim in this paper that electrostatic bonding is preferable to covalent bonding? Could the authors clarify this? Could the authors clearly identify some of the drawbacks of the utilization of this technique (I suppose there are some)?

Answer: We would like to thank the reviewer for her/his comment, as it allows us to better clarify pros and cons of LbL technology for biosensing using nanostructured materials.

We agree with the reviewer that covalent bonding is usually reported to be stronger than electrostatic bonding in many applications, though both strength and stability of the covalent bonding significantly depend on atomic species specifically involved in the bonding (e.g. Si-C [i], Au-S [ii] are very strong, yet Si-O-Si [iii] is not), as well as on experimental conditions (e.g. Si-C are very stable regardless of pH [i], whereas Si-O-Si [iii] is subjected to hydrolysis in water based solutions) required from the specific application.

However, it is important to remark that our main claim here is not specifically related to bonding strength, but it is a more general claim.

In fact, we claim that layer-by-layer nanoassembly of charged polyelectrolytes that were formerly engineered with bioreceptors represents a more effective biofunctionalization route with respect to direct covalent bonding for the preparation of label-free affinity biosensors exploiting nanostructured materials (e.g. silicon-based materials).

In fact, biofunctionalization consists of two chief steps, namely, physio-chemical surface activation and bioreceptor immobilization, both of which have a tremendous effect on selectivity and sensitivity of the resulting biosensor. Yield and stability of the different chemical sub-steps of both surface activation and bioreceptor immobilization processes regulates the number of bioreceptors available at the transducer surface for unit area (bioreceptor density) over time for the biorecognition of the target analyte. Besides, the bioreceptor orientation might also play a role, particularly for affinity biosensing. Moreover, the density of bioreceptors available at the transducers surface is set by both yield and number of the chemical steps needed to activate the surface and secure the bonding of the bioreceptor molecules, being the yield value always <1 (i.e. <100%) for real processes. Therefore, the yield of the entire biofunctionalization process can be relatively low already on flat surfaces as the number of chemical steps increases, and it is expected to be

significantly lower on nanostructured surfaces, where issues related to diffusion, steric demand, and orientation of molecules plays a major role.

In this framework, as pointed out in the manuscript, LbL nanoassembly offers a number of advantages for the biofunctionalization of nanostructured materials, e.g. silicon-based PSi interferometers, compared to covalent bonding. For instance:

1) LbL ensures electrically-induced self-regulation of the thickness of both single and multilayer films, which turns into a superior homogeneity and reproducibility of LbL films when deposited within high-aspect ratio nanostructures [3].

2) LbL ensures good adhesion of the nanolayer to the underlying substrate, because polymers are able to bridge over the underlying defects on the transducer surface (especially for high aspect ratio nanostructured surfaces) [4];

3) LbL assembly form soft and nanoporous surfaces which are advantageous for biocoupling in respect to conserve the structure and activity of the biomolecules, as well as to achieve high loading amounts compared to hard and stiff surfaces of silica or polymers (which are often connected with spacer functions before the biomolecules can be coupled).

4) covalent chemistry biofunctionalization (either silanization or hydrosilylation) is restricted to certain classes of organic linkers that give rise to self-assembled multilayer architectures, e.g. organo-linker/linking-molecule/bioreceptor, of poor quality and with scarce reliability;

5) LbL allows various building-blocks, e.g. the bioreceptors, to be incorporated in individual films and difficult to control (unreliable) chemistry steps, e.g. coupling of bioreceptors to the polymer chains, can be carried out in solution, before going to a nanostructured surface;

6) with LbL assembly the multilayer architecture is completely determined by the deposition sequence, and it is almost independent of the substrate, leading to a controlled and uniform distribution of bioreceptors over the coated surface.

On the other hand, main drawbacks of LbL biofunctionalization of nanostructured materials via electrostatic interaction using charged polyelectrolytes are: 1) unspecific electrostatic interaction of interfering biomolecules with the charged polymer coatings; 2) stability of the LbL assembly at different ionic strengths and pHs.

Both these issues were taken into account and tackled in this research work by:

1) setting up a repulsive rinsing step using a buffer (i.e 10 mM HEPES buffer, pH=7.4, 100 mM NaCl) in this work) with a pH different from the running buffer (i.e. acetate) and greater than pI of all the charged species, so as to exploit electrostatic repulsion of biomolecules unspecifically bound to the polymer (see Fig. 2f);

2) It is true that LbL assembly can have less stability than covalent bonds, though this depends strongly on the polymer combination and working conditions (e.g. pH, ionic strength).

For instance, polyelectrolyte combinations such as PAH or PMAA can be disassembled at extreme pH values (e.g., roughly pH<3 or pH>10) due to protonation or deprotonation or at high ion strength (e.g. above 1- 2 M). Nonetheless, such conditions are not generally applied in biosensing. To this aim, we have thoroughly investigated the stability of LbL nanoassembly in different buffers featuring different pHs and ionic strengths. This allowed acetate (i.e. 10 mM acetate buffer, pH=5, 100 mM NaCl) to be selected as running buffer while HEPES buffer to be selected as repulsive rinsing buffer (see Figure 2b).

However, a further major issue to deal with is reliability and stability of the LbL nanoassembly in a real sample, which contains many different interfering biomolecules.

Triggered by the reviewer's suggestion, we carried out further experiments to test both stability and biosensing performance of PSi interferometers biofunctionalized with LbL nanoassembly in a complex matrix (i.e. whole saliva) spiked 1:10 with acetate buffer at different streptavidin concentrations (namely, 5, 50, and 500 $\mu\text{g/ml}$, i.e. 83, 830, and 8300 nM respectively). Experimental results using saliva show that:

1) upon three injections of diluted saliva without streptavidin (adsorption/repulsive rinsing cycles) a -4.1 ± 5.6 nm was recorded, confirming a good stability of the LbL assembly. Indeed, the ΔEOT value is not statistically different (t-Student confidence level =99%) from that recorded injecting only acetate buffer over 80 min ($\Delta\text{EOT}=-3.8\pm 12.6$ nm, Figure 3b);

2) interfering proteins contained in saliva (namely, enzymes, IgG, IgA, antimicrobial factors, mucosal glycoproteins, albumin, polypeptides, oligopeptides) or small organic molecules derived by metabolic activity [5], although are unspecifically adsorbed onto the LbL charged polyelectrolytes upon saliva injection, they are massively removed during the repulsive rinsing in HEPES (IAW-IAW₀ unspecific signal decreases of roughly 80% after rinsing);

3) biosensing measurements in saliva spiked with different streptavidin concentrations result in output signals (IAW-IAW₀) comparable to those achieved in acetate buffer (Figure 3f-h), further confirming both stability of the LbL assembly in a real body fluid and reliability of the repulsive rinsing step, beyond retainment of the biosensing performance of LbL biofunctionalized PSi interferometers in a complex matrix.

A novel section, with novel figures, entitled *Assessment of LbL biofunctionalized PSi interferometers in saliva*, reporting and discussing novel experiments performed and results achieved on saliva samples was added to the revised version of the manuscript.

Figures (Figure 4f-h and Figure S6a,b)) summarizing experimental results with saliva are reported here below for the reviewer's convenience:

Comment: The authors present a simple case on biotin-streptavidin affinity of the device showing very interesting and promising results. However, as the authors indicate in the manuscript, streptavidin-biotin bond is highly stable and it would be interesting if the authors could test some other bindings, such as IgG (used commonly as a gold standard in different sensing platforms) or comment the major drawbacks to perform other affinity assays using current approach, such as obtaining the adequate polymer or the right orientation for a correct binding.

Answer: We would like to thank the reviewer for her/his interesting comment. Streptavidin/biotin pair (as IgG/anti-IgG pair) is commonly employed as a model for the development of new biosensing either platforms or approaches aimed at the assessment of their analytical performance. A major difference between these two pairs relies on molecular weight and gyration radius of biotin and IgG molecules. In this work, the pair biotin/streptavidin was preferred to IgG/anti-IgG pair due to both lower molecular weight (244.31 Da) and smaller radius of gyration ($\sim 3.55\text{\AA}$) of biotin [6] compared to a conventional IgG molecule ($\sim 150\text{ KDa}$, $\sim 15\text{ nm}$ diameter). In fact, this allows the room left between adjacent biotins in the polyelectrolytes to be augmented, thus resulting in a worst-case test for repulsive rinsing. For instance, taking the polyelectrolyte with biotinylation degree of 1:30 into account and given the distance of a C-C single bond of 1.54 \AA [7], under the hypothesis that biotin is uniformly distributed within the polyelectrolyte, there is a distance of approximately $\sim 9\text{ nm}$ between adjacent biotins, which leave enough room for the unspecific electrostatic interaction of interfering proteins.

Concerning the possibility of performing other assay using LbL biofunctionalization, for example, DNA/c-DNA hybridization assay could be also carried out using LbL nanoassembly once the polyelectrolytes are engineered with a DNA strand linked to the polymer chain. Moreover, the streptavidin/biotin pair itself could be also exploited as first building block to carry out other biofunctionalization assays, such as, for example, using biotinylated bioreceptors (DNA or antibodies) to achieve the proper orientation on the charged polymer.

Following the reviewer comment, the following sentence was added in the conclusions of the revised version of the manuscript:

“On the other hand, further development of the LbL biofunctionalization route both in terms of engineering polyelectrolytes with different bioreceptors, e.g. DNA strands, and using

different/stronger polyelectrolytes, e.g. PSS (pKa=1), is foreseen, which would significantly broaden flexibility and applications of LbL biofunctionalizations.”

Comment: In addition there are some questions that need to be addressed prior to publication:

1.- It would be interesting if the authors could describe schematically the structure formed onto the PSi substrate. Does it consist of a single PAH and B-PMAA layer? Does it have several bilayers?

Answer: We would like to thank the reviewer for pointing this issue out. The LbL assembly used in this work consists of a single PAH layer and a single b-PMAA layer, as sketched in Figure 1a.

Following the reviewer comment, the following sentence in the section *Biofunctionalization of Porous Silicon Interferometers via LbL-assembly* was modified to make clear that a single bi-layer of PAH/b-PMAA is used:

“...results in the conformal electrostatic adsorption of a b-PMAA nanolayer onto the PAH, thus resulting in a single bi-layer of PAH/b-PMAA with bioreceptors directly bonded to the external PMAA polyelectrolyte (Figure 2a-3).”

Comment: 2.- Could you please indicate the parameters for the fabrication of the structures (pH of the solutions, concentrations, temperature, etc.).

Answer: Thank you for pointing this issue out. Although details on the fabrication of both PSi interferometers and LbL assembly were fully provided in the Materials and Methods section of the manuscript (namely, *Materials and chemicals; Preparation and oxidation of PSi interferometers*), we realized that the preparation temperature (18 °C, now added in the revised version of the manuscript) was not provided.

In particular the PSi interferometers were prepared by anodic etching at 600 mA/cm² for 25 s of silicon wafer piece (1.5 x 1.5 cm) with a solution of HF (48%):EtOH, 3:1 v/v, at 18°C. The LbL assembly was instead performed at 18°C using PAH, b-PMAA, and PMAA (1 mg/ml) solubilized in sodium acetate buffer (50 mM CH₃COONa) with the addition of 0.2 M NaCl and adjusted to a pH = 5.6.

The temperature for both PSi interferometer and LbL nanoassembly preparation was now added in the revised version of the manuscript.

Comment: 3.- Could you please comment on the effects of the variation of the fabrication parameters on the final properties of the structure and the implications on the sensing performance of the final device

Answer: In principle, both PSi interferometer preparation and LbL biofunctionalization could affect the biosensing performance of the final device. As described both in the manuscript and in the Supporting Informations to the manuscript, we have thoroughly investigated both these aspects.

Concerning PSi interferometers, the main parameters influencing the sensing performance of the final device are size and thickness of the nanopores.

In fact, both these parameters play a major role in the diffusion of biomolecules inside the PSi interferometer. In particular, the larger the pore size the faster the diffusion kinetics inside the pores; the thinner the pore thickness the shorter the diffusion time throughout the whole pores. As a consequence, for a given biomolecule concentration, larger and shorter pores allow a higher number of biomolecules to be available on the pore surface (i.e. higher density), increasing, in turn, the sensitivity of the device.

A statistical analysis about fabrication of PSi interferometers (over 8 samples) was carried out and reported in Figure S1 (Supplementary Informations), where average values of both thickness ($4.92 \pm 0.16 \mu\text{m}$, red) and of porosity ($80.1 \pm 0.8\%$, blue) of as-prepared PSi interferometers are provided. Very small CV% values are achieved for PSi interferometer fabrication, namely and 3% and 1% for thickness and porosity, respectively, which highlight excellent reliability of PSi interferometer fabrication.

Concerning LbL biofunctionalization, we have thoroughly investigated both reliability of both PAH and b-PMAA depositions (Figure 1d) (n=8) and reproducibility of streptavidin measurements over different samples (n=3) (Figure 3a,b,c), achieving an average CV%=25% over the whole range of streptavidin concentration tested, which is good especially considering the extremely low concentrations tested. Moreover, notice that this CV% value includes variability of PSi interferometer preparation, LbL biofunctionalization, and preparation of the streptavidin solutions in acetate buffer.

Comment: 4.- Could you please comment on the importance of the utilization of HEPES buffer after acetate buffer? Could you please discuss the chemical background of the repulsive behavior of HEPES buffer? Is it a common procedure?

Answer: The use of HEPES buffer (pH=7.4) after acetate buffer (pH=5), that is the repulsive rinsing step, is one of the major original outcomes of this research work and it is actually crucial for the success of LbL biofunctionalization in high-sensitivity biosensing.

In fact, the use of HEPES allows a massive removal of proteins unspecifically adsorbed onto the negatively charged PMAA layer to be achieved through electrostatic repulsion (Figure 2f), this latter occurring when the pH is increased over the isoelectric point (pI) of proteins unspecifically adsorbed on PMAA.

Concerning the chemical background behind the repulsive rinsing, it happens that streptavidin, pepsin, and BSA have a pI of 5, ~1 and 4.7, respectively. Therefore, in acetate buffer (pH=5) they are respectively neutral, negative, and almost neutral. When using HEPES (pH=7.4) all these proteins becomes strongly negatively charged (pH>pI), so that electrostatic repulsion between negatively charged proteins and negatively charged PMAA promotes removal of the formers, if unspecifically bound to PMAA chains. Both HEPES and PBS buffer could be used to obtain a pH=7.4 (<https://www.sigmaaldrich.com/life-science/core-bioreagents/biological-buffers/learning-center/buffer-reference-center.html>). Nonetheless, as discussed in the manuscript (Figure 3b and S3a,b), HEPES was preferred to PBS buffer for the lower ionic strength (at the same molar concentration), which ensures a better stability of the LbL assembly.

During the revision of the manuscript, both reliability and efficacy of the repulsive rinsing in HEPES were successfully tested also on a real body fluid, injecting saliva samples (diluted 1:10 in acetate buffer and spiked with streptavidin) on LbL biofunctionalized PSi interferometers. Saliva is a complex real matrix rich of electrolytes (0.1%), proteins (0.3%) and small organic molecules derived from metabolic activity (0.1%), as reported in [5] Figure S6b shows that, after the repulsive rinsing step, the IAW-IAW₀ unspecific signal due to unspecific adsorption on PMAA of the biomolecules present in saliva is reduced of 80%, (from 0.5 to 0.09 a.u., approximately), in agreement with data achieved in acetate buffer for Pepsin and BSA at similar concentrations (Figure 2f).

To the best of our knowledge this is the first study in which the repulsive rinsing is employed for the desorption of unspecific proteins in biosensing applications.

Comment: 5.- PAH and PAA have been shown as pH sensitive polymers. Could you please indicate the problems associated to this behavior on the sensing performance of the device?

Answer: LbL assembly was carried out at pH=5.6 for both PAH and PMAA polymers. Sensitivity of PAH and PMAA to pH changes could lead to contraction/swelling of the polymers when a change from acetate (pH=5) to HEPES buffer (pH=7.4) occurs.

However, it is important to remind that, biosensing measurements are always made in the same buffer, namely acetate buffer, by measuring changes in the effective refractive index of the sensing architecture induced by bioreceptor/target binding events (namely, difference between IAW value measure in acetate buffer after the binding event and IAW₀ in acetate buffer before the binding event). Therefore, conformational changes of the polymers in buffer with different pH values do not impact on the sensing performance of the device.

Moreover, generally speaking, sensing performance of the device should not be affected by contraction/swelling of polymers with pH changes (for example from acetate buffer, pH=5, to HEPES buffer, pH=7.4). In fact, the biosensor proposed in this work is label-free, that is it is based on the monitoring on effective refractive index changes produced by binding events occurring between bioreceptor and target analytes. Both swelling and contraction of the polymers upon pH changes do not produce any changes in the effective refractive index of the medium, in agreement to the effective medium approximation theory [8]. In fact, the volume percentage of the different materials assembling the sensor architecture, namely silicon, buffer, PAH, and PMAA, does not change upon contraction/swelling of the polymers, and so does the effective refractive index of the sensor.

On the other hand, changes in the pH values of buffers used to carry out the bioassay could lead to LbL disassembling, also triggered by polymer swelling/contraction, which would indirectly affect the sensing performance of the device. To rule this issue out, we thoroughly investigated stability of LbL biofunctionalization in the different buffers used to carry out the bioassay (Figure 3a,b). Experimental results show that the LbL assembly is highly stable both in acetate buffer and in HEPES buffer. In particular, Figure 3a highlights the rapid recovery of the baseline value in acetate buffer after HEPES injection, which neatly confirms that

eventual conformational changes of the LbL assembly (e.g. shrinking or swelling) due to the pH changes are either negligible or rapidly recovered.

Comment: 6.- PAH and PAA are weak polyelectrolytes and their structure can easily change during fabrication if pH is not adjusted properly giving rise to undesired structural properties. How do the authors deal with this problem?

Following the previous comment. Would it be possible to replace PAH and PAA by other strong polyelectrolytes, such as PEI and PSS in order to obtain a more robust device and more repetitive results?

Answer: pKa values of PAH and PMAA are 8.5 [9] and 4.8 [10], respectively.

The ratio between charged polymers and uncharged polymers at pH=5.6 (LbL assembly in acetate buffer) can be estimated accordingly to the Henderson-Hasselbalch equations:

$$pH = pKa + \log \frac{[A^-]}{[HA]} \text{ (for acid)} \quad pH = pKa + \log \frac{[B]}{[BH^+]} \text{ (for base)}$$

For PMAA at pH=5.6, the $\frac{PMAA_{deprotonated}}{PMAA_{protonated}} = 10^{0.8} \sim 6$, meaning that PMAA polymer is sufficiently deprotonated (i.e. negatively charged) during the LbL assembly.

For PAH the ratio between $\frac{PAH_{protonated}}{PAH_{deprotonated}} = 10^{2.9} \sim 800$, meaning that PAH polymer is strongly protonated (i.e. positively charged) during the LbL assembly.

Although PMAA is only slightly deprotonated at pH=5.6, the stability of the whole LbL assembly during the bioassay protocol was experimentally shown to be very good, as also discussed in our answer to comment 6.

Concerning the possible use of other polyelectrolytes, pKa value of PEI is ~ 7 [11], which means that at pH=5.6, PEI polymer is less protonated (positively charged) than PAH. As to PSS, we agree with the reviewer that this polyelectrolyte could lead to a more robust assembly. Indeed, having PSS a pKa=1 [12], by assembling the LbL structure at pH=5.6 the ratio $\frac{PSS_{deprotonated}}{PSS_{protonated}}$ is $10^{4.6} \sim 40000$, which means that the polymer is strongly negatively charged, significantly more than PMAA.

In spite of this, PMAA was preferred due PSS to the presence of carboxylic groups, which can be easily activated by the common NHS/EDAC chemistry for bioreceptor binding.

Following the reviewer suggestion, we decided to comment in the conclusion section of the revised manuscript about the possibility of using other polyelectrolytes (e.g. PSS) to carry out LbL biofunctionalization on materials, so as to make clear the versatility of the proposed approach for biosensing:

“On the other hand, further development of the LbL biofunctionalization route both in terms of engineering polyelectrolytes with different bioreceptors, e.g. DNA strands, and using different/stronger polyelectrolytes, e.g. PSS (pKa=1), is foreseen, which would significantly broaden flexibility and applications of LbL biofunctionalizations.”

Comment: 7.- On Figure S4b) there is a signal shift around minute 70. Could you please comment?

Answer: We would like to thank the reviewer for pointing this issue out. The signal shift recorded in Fig. S4b around minute 70 is an artefact generated by some instrumental temporary issue, probably due to the presence of a microbubble flowing over the sensor for some time during streptavidin injection. Nonetheless, it does not affect the validity of the experimental result.

Reviewer #2

Comment: The authors show a proof-of-concept of the advantages of LbL biofunctionalization, by using nanostructured PSi interferometer, which is “biofunctionalized” for the affinity detection of streptavidin via LbL nano-assembling of PAH with b-PMAA. The interferometers results to be very effective and highly robust, with a detection limit of 600 fM.

It is an advance in the field of diagnose. The claims are novel and they could be of interest to others in the community and the wider field, including chemical biology, medicine and biosensor chemistry.

Answer: We would like to thank the reviewer for her/his positive feedback on our work and for considering it important for a broad audience.

Comment: However, please consider the following points:

1 - LbL technique is very useful to create interfaces for diagnoses. In the introduction part the authors should provide a more deeply and concise information in how does LbL can solve the problem in high-specificity detection of streptavidin. In the context of nanomaterials for diagnosis, what is the challenges and applications in smart devices based on molecular recognition?

Answer: Among the main challenges of biosensors based on molecular recognition, there are stability, specificity, reproducibility, and detection limit.

We showed in the manuscript, and have already discussed in several points in this letter, that experimental data on LbL biofunctionalized PSi interferometers highlight an excellent stability both in buffers (i.e. acetate, HEPES) and real biofluid (i.e. saliva), with an impressive detection limit (600 fM) and a good reproducibility (CV%= 25% over the whole calibration curve).

As to selectivity, it is useful to remind here that, being the outer layer of our LbL assembly negatively charged (PMAA, with deprotonated carboxylic group at the working pH conditions), the selectivity in streptavidin detection compared to unspecific adsorption of streptavidin itself and other proteins (e.g. Pepsin, BSA, proteins in saliva) can be effectively tackled by using a repulsive rinsing step with HEPES buffer (pH=7.4>pI of all proteins). In this condition (i.e. pH>pI) proteins became negatively charged and they are massively desorbed from the surface by electrostatic repulsion.

Following the reviewer suggestion, a more deeply and concise information in how LbL can solve problems in high specificity detection is given in the introduction, as here below reported:

“This is achieved by setting up a repulsive rinsing step at a pH value significantly different from pI values of both target and non-target proteins, so as to make all proteins charged with the same polarity of the outer layer of the LbL nano-assembly.”

However, if there are proteins having a pI larger than 7-8, they will also bind unspecifically to the transducer surface. In that case there are two possible solutions:

1) using additional blocking agents, like PEG or albumin.

2) assembling a LbL structure stable at a pH greater than 7 (e.g. using stronger polyelectrolytes and/or carrying out the LbL at a pH greater than 7).

However, as discussed in the manuscript, working at pH values extensively greater than 7 is not recommended with PSi since chemical stability can be compromised.

Comment: 2- After LBL approach, the stability of the film must be evaluated in a real biological media, for example, blood serum, cellular fluid, among other biological fluids.

Answer: We would like to thank the reviewer for her/his comment. Triggered by the reviewer’s suggestion, we carried out further experiments to test both stability feature and biosensing performance of PSi interferometers biofunctionalized with LbL nanoassembly in a complex biological fluid, namely whole saliva, spiked 1:10 with acetate buffer at different streptavidin concentrations (namely, 83, 830 and 8300 nM corresponding respectively to 5, 50, and 500 µg/ml).

Experimental results on saliva are very encouraging and show that:

1) upon three injections of saliva without streptavidin (adsorption/repulsive rinsing cycles) a $\Delta EOT = -4.1 \pm 5.6$ nm was recorded, which is statistically (t-Student test) equal to that measured upon injection of acetate buffer over 80 min ($\Delta EOT = 3.8 \pm 12.6$ nm), thus confirming a good stability of the LbL assembly in saliva (Figure S6a);

2) interfering proteins contained in saliva (namely, enzymes, igG, IgA, antimicrobial factors, mucosal glycoproteins, albumin, polypeptides, oligopeptides) at concentration of about 300 µg/ml (after dilution), though unspecifically adsorbed onto the LbL charged polyelectrolytes upon saliva injection, are effectively removed during the repulsive rinsing in HEPES, leading to a IAW-IAW₀ signal decrease roughly of the 80%, with an residual IAW-IAW₀=0.090±0.013 a.u. that is consistent to that achieved in acetate buffer for Pepsin and BSA (Figure S6b);

3) biosensing measurements at different streptavidin concentrations result in output signals (IAW-IAW₀) comparable to those achieved in acetate buffer (Figure 3f-h), further confirming both stability of the LbL assembly in real matrix and reliability of the repulsive rinsing step, beyond retainment of the biosensing performance of LbL biofunctionalized PSi interferometers in a real biological fluid.

A novel section, with novel figures, entitled *Assessment of LbL biofunctionalized PSi interferometers in saliva*, reporting and discussing novel experiments performed and results achieved on saliva samples was added to the revised version of the manuscript.

Figures (Figure 4f-h and Figure S6a,b)) summarizing experimental results with saliva are reported here below for the reviewer's convenience:

Comment: 3 – About interference adsorption, could the authors provide a quantitative data?

Answer: Quantitative data on unspecific adsorption of interfering proteins, namely pepsin and BSA, were already provided in Figure 3g, where it is clear that specific binding of streptavidin gives rise to a signal that is significantly higher than those resulting from interference proteins. Remarkably, further experiments carried out during the revision of the manuscript with whole saliva, clearly corroborate the validity on the repulsing rising on the removal of interfering proteins.

In particular Pepsin is almost fully removed from PMAA (being the IAW-IAW₀ recorded at 8.3 μM comparable to the that recorded for the binding of 8.3 pM of streptavidin on b-PMAA, namely 0.033 ± 0.014 a.u.). The sensorgram recorded for BSA (Figure S3d) clearly shows that the removal of BSA unspecifically adsorbed on PMAA is nearly 60% (IAW-IAW₀ = 0.401 ± 0.064 a.u.), which is roughly the same of that achieved using a denaturation solution of 0.5% SDS in aqueous solution as reported in the manuscript.

Comment: 4 – Statistic data should present SD, error and it can be used in the 99% of confidence level in terms of t-Student confidence level. A bunch of data in this regard can give us a more realistic information. Please, consider appropriateness and validity of deeply

statistical analysis, as well the high level of detail must be provided.

Answer: All the relevant experiments assessing the biosensing validity and stability of LbL biofunctionalization (Figure 2 and 3), as well as biosensing performance of LbL biofunctionalized PSi interferometers (Figure 4) were carried out in triplicates (at least), so as to have error bars representing standard deviation values.

Comment: 5 – I wonder to see if it is possible make the miniaturization of the Psi. How small it is possible? Can the authors show us some experiment and result about detector and device, with good stability and reproducibility in micro or sub-micro scale? Please comment it and if it is possible, show some results.

Answer: We thank you the reviewer for her/his interesting comments. As described in the manuscript, PSi is nanostructured materials prepared by electrochemical partial erosion of a silicon substrate. The resulting structure consists of pores with average dimension of 50 nm (in our case) running straight into the silicon substrate. Crystalline silicon walls with thickness in the tens of nanometer scale separate adjacent pores. The degree to which PSi, once prepared, can be patterned to the micrometer and submicrometer, really depends on the patterning tools.

For instance, it has been shown that it is straightforward to pattern PSi down to the submicrometer scale by using both UV lithography approaches [13].

However, a miniaturization of the biosensors presented in this manuscript is out of the scope of this work, which mainly deals with the use of PSi as a difficult-to-functionalize nanostructured materials to both demonstrate and validate the proposed Layer-by-Layer (LbL) biofunctionalization route as a robust and effective alternative approach to standard covalent chemistry.

As a final comment, we would like to point out that the PSi layer prepared in this work is about 1 cm in diameter, whereas the area that is actually probed with optical fiber is only 200 μm (roughly) in diameter.

Comment: 6 – This work is convincing, however, further “real” sample, with no time-consuming techniques to prepare samples would be required to strengthen the conclusions.

Answer: As discussed here above (reviewer 2, comment 2), we successfully carried out further experiments to test both stability feature and biosensing performance of PSi interferometers biofunctionalized with LbL nanoassembly in a complex biological fluid, namely saliva. Remarkably, whole saliva was simply diluted 1:10 with acetate buffer and spiked with different streptavidin concentrations (namely, 5, 50, and 500 $\mu\text{g}/\text{ml}$ i.e. 83, 830 and 8300 nM). So, no time-consuming techniques for the sample preparation were used, but only a simple dilution of whole saliva was used to reduce solid (salts, organic molecules and proteins) to a concentration about 500 $\mu\text{g}/\text{ml}$, as for other analytes studied in this work.

Experimental results on saliva are very encouraging and show that: 1) upon three injections of saliva without streptavidin (adsorption/repulsive rinsing cycles) a $\Delta\text{EOT} = -4.1 \pm 5.6$ nm was recorded, which is statistically (t-Student test) equal to that measured upon injection of acetate buffer over 80 min ($\Delta\text{EOT} = 3.8 \pm 12.6$ nm), thus confirming a good stability of the LbL assembly in saliva (Figure S6a); 2) interfering proteins contained in saliva (namely, enzymes, igG, IgA, antimicrobial factors, mucosal glycoproteins, albumin, polypeptides,

oligopeptides) at concentration of about 300 µg/ml (after dilution), though unspecifically adsorbed onto the LbL charged polyelectrolytes upon saliva injection, are effectively removed during the repulsive rinsing in HEPES, leading to a IAW-IAW₀ signal decrease roughly of the 80%, with an residual IAW-IAW₀=0.090±0.013 a.u. that is consistent to that achieved in acetate buffer for Pepsin and BSA (Figure S6b); 3) biosensing measurements at different streptavidin concentrations result in output signals (IAW-IAW₀) comparable to those achieved in acetate buffer (Figure 3f-h), further confirming both stability of the LbL assembly in real matrix and reliability of the repulsive rinsing step, beyond retainment of the biosensing performance of LbL biofunctionalized PSi interferometers in a real biological fluid.

Comment: 7 - It is true that “bioreceptors can be incorporated in individual films and that difficult to control (unreliable) chemistry steps, e.g. coupling of bioreceptors, can be determined in solution, before going to nanostructured surface.” How does it can be compared with other systems quantitatively? The authors can sustain this information based on real data?

Answer: In this research the label degree of biotin (bioreceptor) on the PMAA polymer was determined in solution with high accuracy before the PMAA is assembled onto the PSi surface. In particular, the determination was carried out by using HABA (4'-hydroxyazobenzene-2-carboxylic acid), a reagent that enables spectrophotometric (colorimetric) measurement of biotinylation levels of labeled proteins or polymers. This assay, which was described in the Materials and Methods of the manuscript, is really straightforward and reliable, and it is well-known since the 1965 [14]. This means that if we assemble always the same amount of PMAA on the PSi surface (as experimentally proven in the manuscript in Fig. 1d), we will have always link to the surface the same amount of available biotin binding sites.

Conversely, to the best of our knowledge, after surficial and covalent coupling between e.g. a biotin-succinimide and an APTES-modified silica surface, it is not easy to measure how many biotin molecules are bound to the surface itself and how many remaining amino-groups there are.

Comment: 8 – The title: “Layer-by-Layer Nano-Assembly as a New Biofunctionalization Route For High-Sensitivity and High-Specificity Label-Free Affinity Biosensing with Nanostructured Materials” sounds very general if we compare it with conclusion. It can be fixed. Also, the “new” should be avoided.

Answer: The title of the manuscript was changed, as per reviewer suggestion, to “Layer-by-Layer Nano-Assembly as a Biofunctionalization Route For High-Sensitivity and High-Specificity Label-Free Affinity Biosensing with Nanostructured Materials: The Case of Nanostructured Porous Silicon”.

Reference

- [1] Cao, X., Ye, Y. & Liu, S. Gold nanoparticle-based signal amplification for biosensing. Anal. Biochem. 417, 1–16 (2011)
- [2] Zhang, S., Garcia-D'Angeli, A., Brennan, J. P. & Huo, Q. Predicting detection limits of enzyme-linked immunosorbent assay (ELISA) and bioanalytical techniques in general. Analyst 139, 439–445 (2014).

- [3] Dähne, L.; Baude, B. Method for producing CS particles and microcapsules using porous templates, cs particles and microcapsules, and the use thereof (2004) Az102004013637.8.
- [4] Dähne, L.; Peyratout, C. Nanoengineered capsules with Specific Layer Structures in Review in Dekker Encyclopedia of Nanoscience and Nanotechnology 2004, 2355-2367.
- [5] Kumar, B. et al. The composition, function and role of saliva in maintaining oral health : A review. International Journal of Contemporary Dental and Medical Reviews. 2017, 011217, 1–6.
- [6] Lei, Y., Li, H., Zhang, R. & Han, S. Molecular Dynamics Simulations of Biotin in Aqueous Solution. J. Phys. Chem. B 108, 10131–10137 (2004)
- [7] Pauling, L. & Brockway, L. O. Carbon—Carbon Bond Distances. The Electron Diffraction Investigation of Ethane, Propane, Isobutane, Neopentane, Cyclopropane, Cyclopentane, Cyclohexane, Allene, Ethylene, Isobutene, Tetramethylethylene, Mesitylene, and Hexamethylbenzene. Revised Values of Covalent Radii. J. Am. Chem. Soc. 59, 1223–1236 (1937).
- [8] Sailor, M. J. (2012). Characterization of Porous Silicon. In Porous Silicon in Practice, M. J. Sailor (Ed.). doi:10.1002/9783527641901.ch5. Page 153.
- [9] (Cranford, S. W., Ortiz, C. & Buehler, M. J. Mechanomutable properties of a PAA/PAH polyelectrolyte complex: rate dependence and ionization effects on tunable adhesion strength. Soft Matter 6, 4175–4188 (2010));
- [10] Peppas, N., Hilt, J. Z., Thomas, J. B. Nanotechnology in Therapeutics: Current Technology and Applications, Taylor & Francis, (2007)).
- [11] Curtis, K. A. et al. Unusual Salt and pH Induced Changes in Polyethylenimine Solutions. PLoS One 11, e0158147 (2016)
- [12] (Lewis, S. R. et al. Reactive nanostructured membranes for water purification. Proc. Natl. Acad. Sci. 108, 8577 LP-8582 (2011))
- [13] Cunin, F. et al. Biomolecular screening with encoded porous-silicon photonic crystals. Nat. Mater. 1, 39 (2002)) and direct imprinting (J.D. Ryckman et al, Nano Lett. (2011))
- [14] Green, N. M. A Spectrophotometric Assay for Avidin and Biotin Based on Binding of Dyes by Avidin. Biochem. J. 94, 23C LP-24C (1965).

REVIEWERS' COMMENTS:

Reviewer #2 (Remarks to the Author):

This article had its content improved and all my questions were answered pertinently. I recommend the publication as revised.